# DECOMPOSE THE MODEL: MECHANISTIC INTERPRETABILITY IN IMAGE MODELS WITH GENERALIZED INTEGRATED GRADIENTS (GIG)

## ABSTRACT

In the field of eXplainable AI (XAI) in language models, the progression from local explanations of individual decisions to global explanations with high-level concepts has laid the groundwork for mechanistic interpretability, which aims to decode the exact operations. However, this paradigm has not been adequately explored in image models, where existing methods have primarily focused on class-specific interpretations. This paper introduces a novel approach to systematically trace the entire pathway from input through all intermediate layers to the final output within the whole dataset. We utilize Pointwise Feature Vectors (PFVs) and instance-specific Effective Receptive Fields (iERFs) to decompose model embeddings into interpretable Concept Vectors. Then, we calculate the relevance between concept vectors with our Generalized Integrated Gradients (GIG), enabling a comprehensive, dataset-wide analysis of model behavior. We validate our method of concept extraction and concept attribution in both qualitative and quantitative evaluations. Our approach advances the understanding of semantic significance within image models, offering a holistic view of their operational mechanics. [1]

## 1 INTRODUCTION

In the field of eXplainable AI (XAI), efforts have historically transitioned from Local explanation to Global explanation to Mechanistic Interpretability. While local explanation methods including Selvaraju et al. (2017); Montavon et al. (2017); Sundararajan et al. (2017); Han et al. (2024) have focused on explaining specific decisions for individual instances, global explanation methods seek to uncover overall patterns and behaviors applicable across the entire dataset (Wu et al., 2023; Xuanyuan et al., 2023; Singh et al., 2024). One step further, mechanistic interpretability methods seek to analyze the fundamental components of the models and provide a holistic explanation of operational mechanics across various layers.

Recently, researchers in language models, (Geva et al., 2022; Bricken et al., 2023; Gurnee et al., 2024), have extensively studied mechanistic interpretability to reveal the precise mechanisms transforming inputs into outputs. They provide a dataset-wide explanations by utilizing the whole instances from the dataset, regardless of the classes. Interpretability in image models (Fel et al., 2023; Ghorbani et al., 2019), however, have typically focused on class-wise explanations, which interpret model decisions using only data from a specific class, thereby failing to capture shared concepts across different classes. This distinction arises because images consist of pixels that do not inherently represent concepts, unlike languages where each comprising word itself can be treated as a concept. Additionally, as meaningful structures in images are localized and only occupy small regions of the entire image, the embedding space in image datasets is far more sparse compared to that in language datasets.

In this paper, we present a novel approach to mechanistic interpretation in image models by systematically decomposing and tracing the pathways from input to output across an entire dataset. Unlike previous methods that often focus on individual classes or specific features, our approach provides

---

[1]https://iclr2025gig.netlify.app/graph_visualization.html

a comprehensive, dataset-wide understanding of the entire model's behavior. This is the first to explain the model's embedding within the whole dataset, throughout the whole layers (See Fig. 1). We decompose the model's embedding with the dataset-wide concept vectors, enabling the existence of "Shared Concepts" unlike previous class-wise explanation methods (Fel et al., 2023), (Kowal et al., 2024).

Following the framework of Han et al. (2024), we use the Pointwise Feature Vector (PFV) for our analysis unit, which is defined as the channel-axis pre-activation vector of each layer, serving as the fundamental unit encoding the network's representations. We further utilize the instance-specific Effective Receptive Fields (iERFs) of the PFVs to label their semantic meaning, enabling a direct investigation of the PFV vector space. By leveraging iERF, we aim to identify clear and meaningful linear bases within the PFV vector space, allowing us to decompose previously unknown PFVs into interpretable principal components, which we refer to as Concept Vectors (CVs).

To find out the bases and identify the CVs, we leverage several clustering methods including dictionary learning, k-means, and Sparse AutoEncoder. Among the clustering methods, we employ bisecting k-means clustering, since it is well-suited for the PFV vector space, which is highly sparse and variably dense. For instance, background features often cluster densely, while critical features, such as "the beak of a bird", may occupy a broader, less dense area.

To quantify the causal relationships between concept vectors in different layers, we introduce **Generalized Integrated Gradients** (GIG), which effectively captures interlayer contributions. By combining Concept Vectors with GIG, we can offer a comprehensive causal analysis of the ResNet50 model, from the lowest layers to the final class predictions. Fig. 1 shows examples of how our method iteratively aggregates concepts in the previous layer to form a higher-layer abstract concept.

In this study, we focus on the ResNet50 architecture. Yet, our approach is not limited to convolutional architectures and can be universally applicable across various modalities, including transformer architectures, which we intend to explore in future work. Our framework facilitates a deeper understanding of the semantic significance of features, thus advancing the mechanistic interpretability image models.

## 2 RELATED WORKS

**Local explanation Methods**, often referred to as attribution methods, such as Sundararajan et al. (2017); Montavon et al. (2017); Han et al. (2024), aim to explain model predictions for specific instances. These methods attribute the output to particular input features, such as pixels or neurons, offering instance-specific insights. However, they often suffer from reliability issues due to an ambiguity in interpreting what the generated explanation maps actually signify (Adebayo et al., 2018).

To address this, **global explanation methods**, known as concept attribution methods (Kim et al., 2018; Fel et al., 2023), extend traditional attribution methods by associating model predictions with high-level concepts. They aim to provide explanations at a broader, global scale by identifying and quantifying the importance of predefined or automatically extracted concepts. However, these methods remain confined to a single layer of the model and are typically class-specific. This limitation highlights the need for methods that can analyze interactions between concepts across multiple layers.

Lastly, **mechanistic interpretability**, widely recognized as interlayer concept attribution methods (Achtibat et al., 2023; Kowal et al., 2024), take the field a step further by tracing the evolution and interaction of concepts across multiple layers of a model. Unlike local or global explanation methods, which are localized to specific instances or layers, these methods focus on systematically analyzing the model's internal mechanisms throughout its entire architecture. For instance, CRP (Achtibat et al., 2023), an extension of LRP (Bach et al., 2015), provides detailed exploration of how the concepts impact the model's output at each layer, by introducing concept-conditional relevance mapping. VCC (Kowal et al., 2024), an extension of TCAV (Kim et al., 2018), interprets how individual concepts contribute to the model's decisions across different layers. However, both of them focus on a specific class, thereby limiting their capacities to offer a more comprehensive and generalized view of the model's behavior across the entire dataset. In contrast, our approach broadens the interlayer analysis to include the entire dataset, enabling a more thorough examination of the model's decision-making process. Our work is the first to provide dataset-wide mechanistic

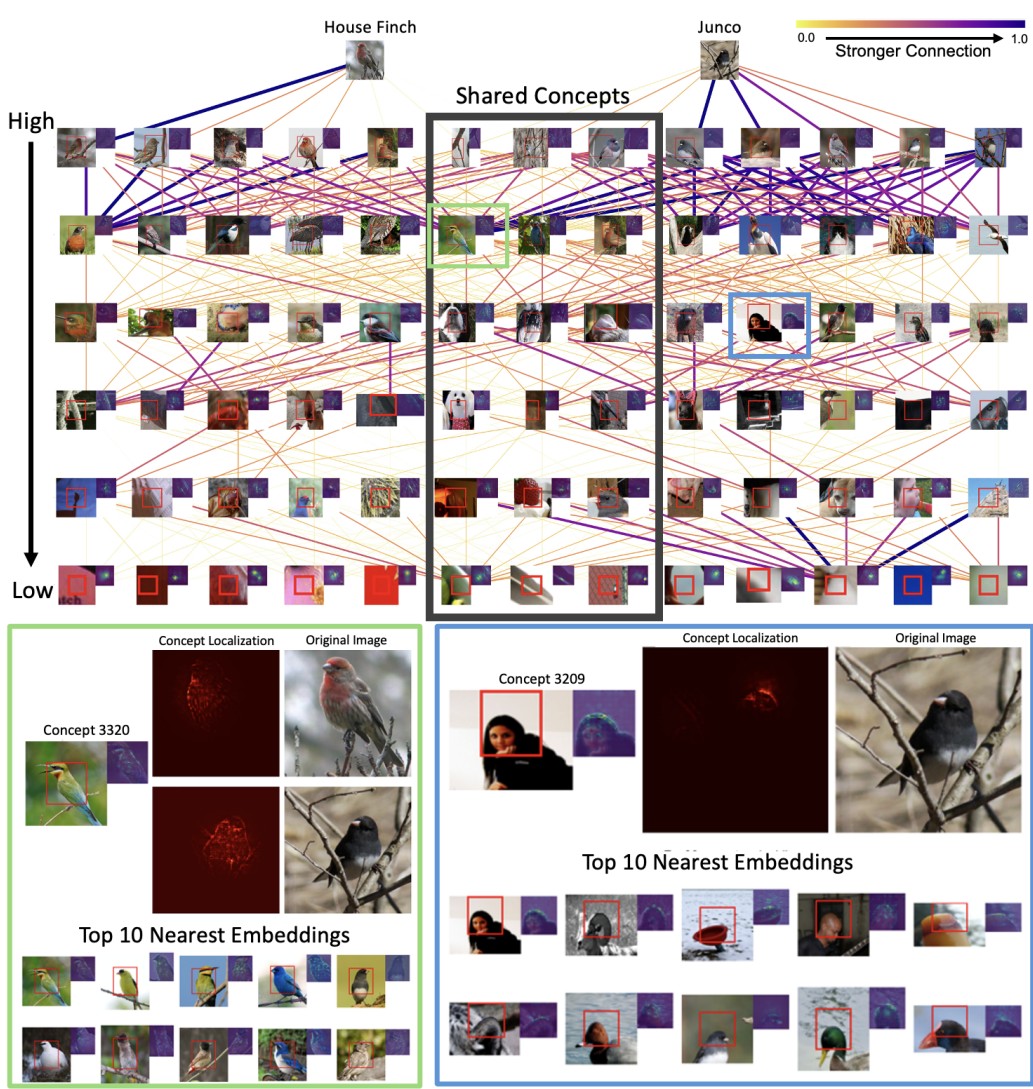

Figure 1: **Top**: Causal explanation graph from high to low layers. From top to bottom, [Classifier, Layer4.2, Layer3.5, Layer3.2, Layer3.0, Layer2.3, Layer1.2], the bottleneck blocks in ResNet50 (All-layer analysis is provided in Appendix A). The thicker and bluer the edge, the stronger the contribution between concepts. Unlike class-wise global explanation, our method can explain the 'Shared concepts' between similar classes. Among the thousands of concepts in a layer, the graph only shows the top-5 most important concepts and top-3 shared concepts. **Bottom left**: Detailed concept visualization of Concept 3,320 "Bird chest" at Layer3.5 Block. With the top 10 nearest embeddings, we can observe Concept 3,320 is "Bird Torso." With its concept localization image, we can effectively see where concept 3,320 resides in input images of the classes, house finch and junco, respectively. **Bottom right**: Concept 3,209 "Round head" at Layer3.2 Block. The top-1 representation image of concept 3,209 (A girl's head) seems irrelevant to the class, junco bird. However, with the concept localization image and the corresponding top-10 nearest embeddings, we can see that Concept 3,209 represents the round head of various objects.

interpretability by tracing the pathway from input images through all intermediate layers to the final output, uncovering complex interactions that previous works may overlook.

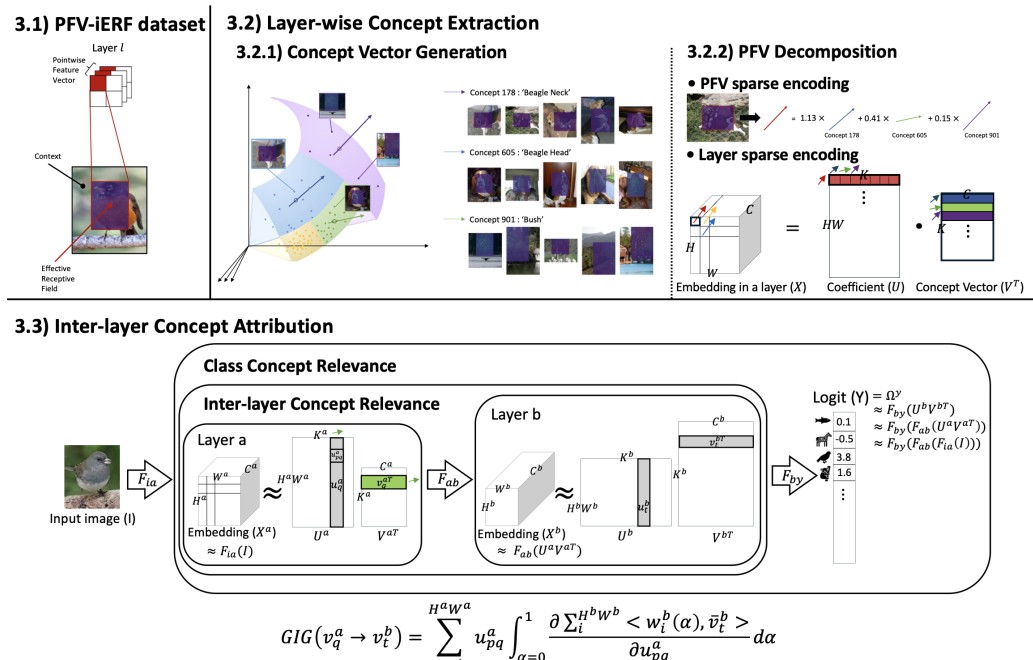

Figure 2: Method overview. **3.1)** Our dataset. The Pointwise Feature Vector (PFV) in the hidden layer is assigned a meaning by labeling it with the instance-specific Effective Receptive Field (iERF). The iERF image (the blue area in the picture) uses a single color to represent the importance, making it difficult to interpret. Therefore, the context portion, around the iERF, is added to make the significance of the iERF more understandable. **3.2)** Layer-wise concept extraction. **3.2.1)** The PFV vector space exhibits a diverse density, with high density around specific concepts and sparsity elsewhere. Hence, bisecting clustering, suitable for such data structures, is employed to extract concept vectors. The meaning of each concept vector is then explained through the sample with the highest cosine similarity to the concept vector. **3.2.2)** Reconstruct the PFV and embeddings in a layer with the extracted concept vectors. **3.3)** Inter-layer concept attribution, employing Generalized Integrated Gradients (GIG).

# 3 METHOD

Fig. 2 shows the overview of method. In the figure, each process of our method is represented with the corresponding section number.

## 3.1 ANALYSIS UNIT: PFV-iERF DATASET

In our study on mechanistic interpretability in image models, we utilize several key components essential for understanding and analyzing the network's behavior from the work of Han et al. (2024). Specifically, we use a Pointwise Feature Vector (PFV) as the unit of analysis and its corresponding instance-specific Effective Receptive Field (iERF) as the visual label to effectively show and validate the knowledge encoded by the PFV.

Firstly, a PFV is a vector of neurons along the channel axis within a hidden layer that share an identical receptive field. Given the embedding of layer $l$ denoted as $A^l \in \mathbb{R}^{H^l W^l \times C^l}$, where $C^l$ is the number of channels and $H^l W^l$ represents the spatial dimensions of the feature map, the PFV at position $p \in \{1, \cdots, H^l W^l\}$ is represented as $\mathbf{x}_p^l \in \mathbb{R}^{C^l}$. This vector encapsulates a localized feature representation at a specific point within the input image, providing a clear characterization of the features at that particular location. Unlike individual neurons, a PFV ensures monosemanticity, capturing a singular, coherent concept from the multi-channel features at a specific spatial

location. Therefore, we decompose a layer in a network using PFVs. More specifically, a PFV in the preactivation space is linearly decomposed with the concept vectors.

Secondly, we use the iERF as the PFV's label. Receptive Field (RF) denotes the region within the input image that influences the activation of a specific feature, defining the spatial extent over which the input pixels contribute to the feature's activation. Luo et al. (2016) introduced the concept of the Effective Receptive Field (ERF), revealing that contrary to the theoretical RF, the actual impact of input pixels is concentrated around the center and follows a Gaussian distribution. Han et al. (2024) further refined this concept to instance-specific Effective Receptive Field (iERF) to highlight the differential impact of individual pixels, identifying those that are most influential in the computation of the PFV. With iERF, we directly attribute a meaning (or a concept) to each hidden layer feature vector (PFV in our case), in contrast to other existing methods, which infer feature vector meaning through indirect techniques. Ghorbani et al. (2019) and Kowal et al. (2024) used global average pooling after masking, and Fel et al. (2023) used bilinear interpolation on the masked feature maps to create a squared region to provide an indirect explanation of feature vectors by transforming segmented areas into representative vectors. Yet, with iERF, we explicitly assign meanings to the hidden layer feature vectors, treating them as representations of specific concepts so that we can offer a more straightforward interpretation of how particular features contribute to the model's decisions.

## 3.2 CONCEPT EXTRACTION

### 3.2.1 CONCEPT VECTOR GENERATION

In our framework, a concept is defined as a linear basis in the high-dimensional PFV space that represents semantically meaningful features across different images, transcending classes. To find out the concept vector in each layer, we utilize ImageNet validation dataset, consisting of 50,000 images. Even though there are $HW$ PFVs within a single layer, we take only one PFV and its corresponding iERF, resulting in 50,000 PFV-iERF pairs per layer for the dataset. In an image, we sample a PFV in a non-uniform sense to reflect its contribution to the output (logit), due to the foreground-background imbalance problem in images; If we sample PFVs randomly from an image, then the majority would capture the background, which would be irrelevant to the output class. For example, the sky in an image could be present across various classes, leading to an overrepresentation of the class-irrelevant feature, sky, rather than critical features like a bird's beak. This overrepresentation of irrelevant features within the PFVs could skew the identification of the principal axes of the PFVs. Thus, to address the foreground-background imbalance, we sample a single Pointwise Feature Vector (PFV) from each image. This PFV is chosen probabilistically, with a preference for those that contribute more significantly to the model's output logits. This approach ensures that the selected PFVs are highly relevant to the class predictions, thereby creating a more balanced representation of important features across the dataset. Details of how the contribution was calculated are provided in the Appendix. C.

With this balanced PFV-iERF dataset, we employ a bisecting k-means clustering (Steinbach et al., 2000), which iteratively splits the data into two clusters until a predefined number of clusters is reached. This approach effectively navigates the complex manifold of image data, where some regions are sparse, containing rare or atypical features, while others are dense, filled with frequently encountered features. After clustering, we assign the centroid of each cluster as a concept vector. The detailed procedures are included in the Appendix. B.

### 3.2.2 PFV DECOMPOSITION

Let there be $k$ concept vectors in layer $l$, denoted as $\mathbf{v}_1^l, \cdots, \mathbf{v}_k^l$, discovered in the same $C$-dimensional vector space $\mathcal{V}^l$ with PFVs. Then, each PFV $\mathbf{x}_p^l$ can be expressed as a linear combination of the concept vectors:

$$\mathbf{x}_p^l = \sum_{j=1}^{k} u_{pj} \mathbf{v}_j^l + \epsilon, \tag{1}$$

where $u_{pj}$ is the coefficient representing the contribution of the $j$-th concept vector to PFV $x_p^l$ ($\mathbf{u}_p = [u_{p1}, \cdots, u_{pk}]^T$), and $\epsilon$ is the residual error. To determine the coefficients $\mathbf{u}_p$, we use Lasso

regression, which minimizes the following objective function:

$$\mathbf{u}_p^* = \arg\min_{\mathbf{u}_p} \left\{ \frac{1}{2} \left\| \mathbf{x}_p^l - \sum_{j=1}^k u_{pj} \mathbf{v}_j^l \right\|_2^2 + \lambda \sum_{j=1}^k |u_{pj}| \right\}, \tag{2}$$

where $\lambda$ is a regularization parameter that controls the sparsity of the solution, encouraging many of the coefficients $u_{pj}$ to be zero. By using lasso regression, we can reconstruct the original PFVs with a small number of concept vectors.

In this way, the embeddings in the $l$-th layer, $X^l \in \mathbb{R}^{HW \times C}$, can be approximated by $k$ concept vectors as $\tilde{X}^l = UV^T$ where $U \in \mathbb{R}^{HW \times k}$ is the coefficient matrix and each column of $V \in \mathbb{R}^{C \times k}$ contains a concept vector. Refer to the Appendix E for specific examples.

### 3.3 INTER-LAYER CONCEPT ATTRIBUTION

In this paper, we leveraged Integrated Gradients (IG) (Sundararajan et al., 2017) to calculate the inter-layer concept attribution. Among other attribution methods, we utilized IG due to its superiority across various reliability metrics, such as C-Deletion, C-Insertion, and C-μFidelity, which are crucial in ensuring the robustness and accuracy of concept-based explanations (Fel et al., 2024).

Based on IG, we propose a novel method, **Generalized Integrated Gradients (GIG)**, which extends the integrated gradients to quantify the contribution of a specific concept vector in a layer to both the final class output and the concept vectors of subsequent layers.

Let $a$ and $b$ denote the preceding and target layer, and $X^l (l \in \{a, b\})$ be the embeddings of the corresponding layer. In this work, we want to measure the influence of a query concept vector in layer $a$, $\mathbf{v}_q^a$, on the target concept vector in layer $b$, $\mathbf{v}_t^b$. To compute the attribution for the target concept vector, we first compute the output embeddings

$$\Omega^b(\alpha) = F_{ab}(\alpha U^a (V^a)^T) \tag{3}$$

in layer $b$ by varying the embeddings in layer $a$ from 0 to $\tilde{X}^a = U^a(V^a)^T$, i.e, $\alpha \in [0, 1]$ in Eq. (3). Here, $F_{ab}$ represents the nonlinear function from layer $a$ to $b$ and $U^a(V^a)^T$ is the approximation of $X^a$ obtained in Sec. 3.2.2. Then, we project $\Omega^b(\alpha)$ onto the target concept vector $\mathbf{v}_t^b$ and obtain the projected vectors. These projected vectors are spatially aggregated and we compute the integrated gradients for the $q$-th element of the coefficient vector, $u_{pq}$, which is the component of $\mathbf{v}_q^a$ in the PFV $\mathbf{x}_p^a$ at position $p$ as follows:

$$\text{GIG}(\mathbf{v}_q^a|_p \to \mathbf{v}_t^b) = u_{pq}^a \int_{\alpha=0}^1 \frac{\partial \sum_{i=1}^{H^b W^b} \langle \mathbf{w}_i^b(\alpha), \bar{\mathbf{v}}_t^b \rangle}{\partial u_{pq}^a} d\alpha. \tag{4}$$

Here, $\bar{\mathbf{v}}_t^b$ is the normalized version of $\mathbf{v}_t^b$, $\langle \cdot, \cdot \rangle$ is the inner product operation and $\mathbf{w}_i^b$ is the embedding of $\Omega^b$ at the $i$-th position.

Note that the above GIG measures the attribution of the query concept vector at position $p$ to the target concept vector in a subsequent layer. To measure the attribution of a query concept vector, $\mathbf{v}_q^a$, to the target concept vector, $\mathbf{v}_t^b$, we sum up all the attributions of $\mathbf{v}_q^a$ at different positions as follows:

$$\text{GIG}(\mathbf{v}_q^a \to \mathbf{v}_t^b) := \sum_{p=1}^{H^a W^a} \text{GIG}(\mathbf{v}_q^a|_p \to \mathbf{v}_t^b) = \sum_{p=1}^{H^a W^a} u_{pq}^a \int_{\alpha=0}^1 \frac{\partial \sum_{i=1}^{H^b W^b} \langle \mathbf{w}_i^b(\alpha), \bar{\mathbf{v}}_t^b \rangle}{\partial u_{pq}^a} d\alpha. \tag{5}$$

**Class Concept Relevance**    To quantify the class importance score of a query concept vector in layer $a$, $\mathbf{v}_q^a$, for the final output (contribution of the concept vector to the given class), we treat each class label $c$ as an independent concept. Thus, we convert the class index into a one-hot vector, $\mathbf{e}_c \in \{0, 1\}^N$, where $N$ is the number of classes:

$$(\mathbf{e}_c)_i = \begin{cases} 1 & \text{if } i = c \\ 0 & \text{if } i \neq c. \end{cases} \tag{6}$$

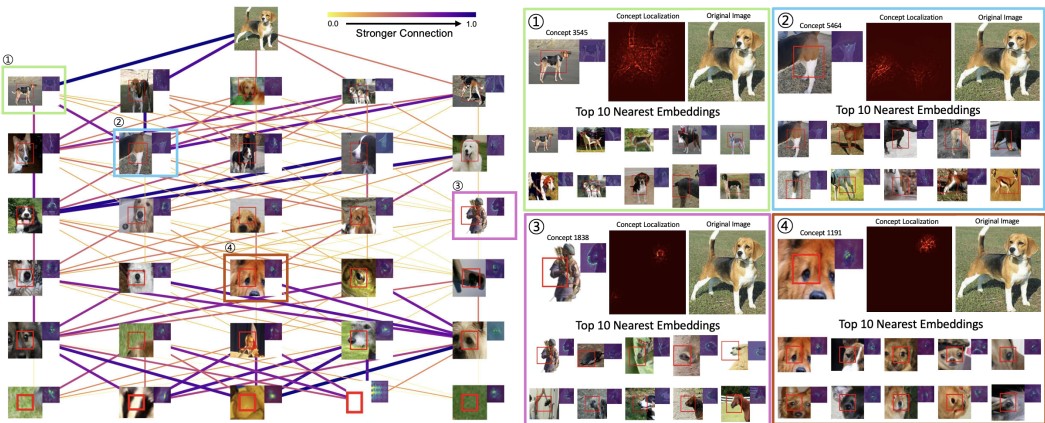

Figure 3: **Left**: Causal explanation graph of 'Foxhound'. From top to bottom, [Classifier, Layer4.2, Layer3.5, Layer3.2, Layer3.0, Layer2.3, Layer1.2], the bottleneck blocks in ResNet50. Our method can also provide a dataset-wide explanation of a single image. **Right**: Detailed concept visualization of the colored boxes, 1) green, 2) blue, 3) purple, and 4) brown. ① Concept 3,545 "Brown-white-black fur Body" at Layer4.2 Block. ② Concept 5,464 "Brown-white-black Thigh" at Layer3.5 Block. ③ Concept 1,838 "Rounded Cone" at Layer3.2 Block. With Top-1 representation image of 'folded arm', the concept seems irrelevant to the input image. However, the concept localization and the top-10 nearest embeddings show that Concept 1,838 represents "Rounded Cone". ④ Concept 1,191 "Eye" at Layer2.3 Block. Best viewed when enlarged.

Then, we calculate the class contribution of the concept vector, $\mathbf{v}_q^a$ using Eq. (5) as follows:

$$\text{GIG}(\mathbf{v}_q^a \to \mathbf{e}_c) := \sum_{p=1}^{H^a W^a} \text{GIG}(\mathbf{v}_q^a|_p \to \mathbf{e}_c) = \sum_{p=1}^{H^a W^a} u_{pq}^a \int_{\alpha=0}^{1} \frac{\partial \langle \mathbf{w}^y(\alpha), \mathbf{e}_c \rangle}{\partial u_{pq}^a} d\alpha, \qquad (7)$$

where $y$ indicates the output layer and $\mathbf{w}^y(\alpha)$ is the predicted class probability vector for input embeddings scaled by $\alpha$, i.e, $\mathbf{w}^y = F_{ay}(\alpha U^a (V^a)^T)$.

By employing our Generalized Integrated Gradients, we aim to uncover the mechanistic interpretability in image models, providing a detailed understanding of how these networks process image data and construct specific concepts through the layers.

## 4 EXPERIMENT

To demonstrate the effectiveness of our method, we provide two kinds of qualitative analysis including one-class explanation (Fig. 3) and two-class explanation (Fig. 1). Furthermore, we validate our method of concept extraction and concept attribution with comprehensive experiments in Sec. 4.2.

**Settings.** Following Bricken et al. (2023), we selected the concept size of each layer as 8 times the number of channels in that layer, making overcomplete linear basis. For classic dictionary learning, we utilized the Least Angle Regression (LARS) algorithm and the Lasso LARS algorithm for PFV decomposition. For sparse autoencoder, we followed the setting of Templeton et al. (2024). For both methods, we extend them by decomposing PFVs directly into coefficients and concept vectors without relying on global average pooling, as they have been applied either at the token level within Transformer architecture, or on the global average pooled outputs of ResNet50 architecture. Due to excessive computational time, we cannot obtain data of higher layers using the class dictionary learning.

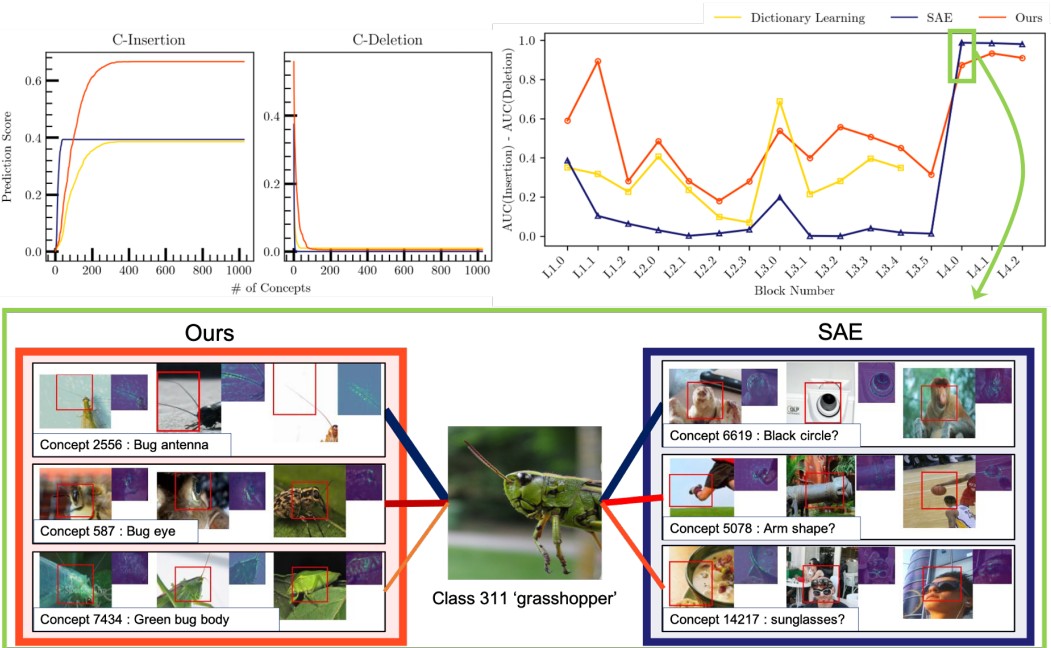

Figure 4: Validation of Concept Extraction. **Top Left**: Comparison of C-Insertion and C-Deletion curves for three concept extraction methods applied to ResNet50's first block of first stage (Layer1.0). **Top Right**: AUC differences across different block numbers for a balanced comparison, as there is a tendency that the better the insertion performance, the worse the deletion performance. **Bottom**: Top 3 most important concepts found by Sparse AutoEncoder (SAE) and 'Ours' for classifying grasshopper image at Layer4.0. Even though SAE excels our method in AUC difference on later layers, the concepts extracted by SAE seem less persuasive than those from 'Ours'.

## 4.1 QUALITATIVE ANALYSIS

As seen in Fig. 3, we can explain how the concept components are constructed through layers. Moreover, as shown in Fig. 1, we can even find out the shared concepts, since we analyze the models within the whole dataset, not a specific class.

## 4.2 VALIDATION OF OUR METHOD

Since our method involves two main steps, we validate the steps of our method with both qualitative and quantitative experiments: Sec. 4.2.1 for Concept Extraction, Sec. 4.2.2 for PFV Decomposition and Sec. 4.2.3 for Inter-layer Concept Attribution.

### 4.2.1 VALIDATION OF CONCEPT EXTRACTION

To validate our method, we assess its fidelity using the C-Deletion and C-Insertion metrics, as proposed by Fel et al. (2024). These methods provide a robust framework for evaluating the alignment between our explanation model and the original model's behavior by systematically modifying concept activations and observing the resulting impact on model predictions.

In C-Deletion and C-Insertions, concept vectors are removed or inserted in the order of their importance, and the Area Under the Curve (AUC) of the accuracy drop graph is measured. The importance score of a concept is calculated with Eq. (7), as it is the most reliable CAT method (Fel et al., 2024). For C-Deletion, a lower AUC indicates a more effective extraction method, as it signifies a greater impact on model performance when key concepts are removed. Conversely, in C-Insertion, a higher AUC is preferable, reflecting a more accurate prediction when important concepts are introduced. Finally, we measure the AUC difference to see the overall trends in every layer.

**Results** As shown in the top part of Fig. 4, 'Ours' with Bisecting Clustering demonstrated consistently strong performance across most layers in both C-Deletion and C-Insertion. Considering the fidelity metric, the difference between AUC(Insertion) and AUC(Deletion), 'Ours' outperforms the other methods, demonstrating its effectiveness in capturing and utilizing the most essential features of the model. We observed that while SparseAutoEncoder (SAE) exhibited lowest fidelity in the earlier layers, it demonstrated exceptional performance in C-Insertion, achieving the highest AUC values in layer 4. However, as seen in the bottom part of Fig. 4, while the concepts from 'Ours' are human-interpretable, those from SAE seem ambiguous and even irrelevant to the class.

### 4.2.2 VALIDATION OF PFV DECOMPOSITION

| Method | | Layers | | | | | | | |
|---|---|---|---|---|---|---|---|---|---|
| | | **Layer1.1** | **Layer2.0** | **Layer2.2** | **Layer3.0** | **Layer3.2** | **Layer3.4** | **Layer4.0** | **Layer4.2** |
| DictionaryLearning | Rel-$l_2$($\downarrow$) | 0.5968 | 0.6103 | 0.7795 | 0.6499 | 0.8235 | 0.7851 | - | **-** |
| | $l_0$ ratio($\uparrow$) | **0.9952** | 0.9958 | 0.9955 | 0.9953 | **0.9976** | **0.9982** | - | - |
| SAE | Rel-$l_2$($\downarrow$) | 1.8952 | 1.5028 | 1.2524 | 1.2617 | 1.5257 | 1.4643 | **0.5096** | **0.4816** |
| | $l_0$ ratio($\uparrow$) | 0.9622 | 0.9787 | 0.9779 | 0.9842 | 0.9889 | 0.9925 | 0.9921 | 0.9913 |
| Ours | Rel-$l_2$($\downarrow$) | **0.3651** | **0.4633** | **0.6159** | **0.5618** | **0.6855** | **0.6565** | 0.6051 | 0.5095 |
| | $l_0$ ratio($\uparrow$) | 0.9947 | **0.9964** | **0.9961** | **0.9971** | 0.9967 | 0.9962 | **0.9968** | **0.9936** |

Table 1: Relative $l_2$ error and $l_0$ ratio for the odd-numbered layers when reconstructing PFVs with each concept extraction methods.

To validate the effectiveness of PFV decomposition, we calculated relative $l_2$ for reconstruction error and $l_0$ ratio for sparsity. Relative $l_2$ error (Rel-$l_2$) is defined as the ratio of the reconstruction error to the $l_2$ norm of the original vector, and $l_0$ ratio is calculated as the ratio of the number of zero coefficients to the total number of concept vectors.

**Results.** From Tab. 1, we observed that our method consistently achieves lower Relative $l_2$ errors across most layers, compared to the baseline methods. Specifically, in blocks from Layer1.1 to Layer3.4, our approach outperformed both classical Dictionary Learning and Sparse AutoEncoder (SAE), indicating more accurate reconstruction of PFVs.

In terms of sparsity, our method maintained competitive levels. While classical Dictionary Learning showed slightly better sparsity in (e.g., Layer1.1 and Layer3.2), it did so at the expense of higher reconstruction errors. SAE, on the other hand, exhibited higher coefficient $l_0$ values, indicating less sparse representations except bottleneck blocks in Layer4.

These results validated that our PFV decomposition effectively captures the essential features of the PFVs using a minimal set of concept vectors, reducing the computational cost (Appendix. H). By outperforming baselines in both reconstruction accuracy and sparsity, our approach demonstrates its potential for efficient representation in high-dimensional spaces.

### 4.2.3 VALIDATION OF INTER-LAYER CONCEPT ATTRIBUTION

To validate the effectiveness of our concept attribution method, Generalized Integrated Gradients (GIG), we adapted the concept insertion and deletion strategies typically used in evaluating Concept ATtribution (CAT) methods. The original C-Insertion and C-Deletion metrics quantify the relationship between the identified concept vectors and the target class. By systematically inserting or deleting concept vectors according to their attribution scores and observing changes in the target class score, we assessed the validity of the concept vectors of CAT methods.

We extended this metric to validate the relationship between concept vectors in different layers. As derived in Sec. 3.3, the class label can be seen as the one-hot concept vector of the last layer after the fully-connected layer. Therefore, we validated the efficacy of our inter-layer concept attribution method, observing the changes in the direction of the target concept vector in the subsequent layer by inserting or deleting concept vectors from a preceding layer. Specifically, we deleted or inserted concept vectors from the source layer one by one and observed the changes in the output of the target layer (with dimensions $H^b \times W^b \times C$). For instance, if we delete concept vectors related to the target concept "dog nose" from the source layer in order of their GIG attribution, the output in the target layer corresponding to the "dog nose" direction should decrease accordingly. However, given that

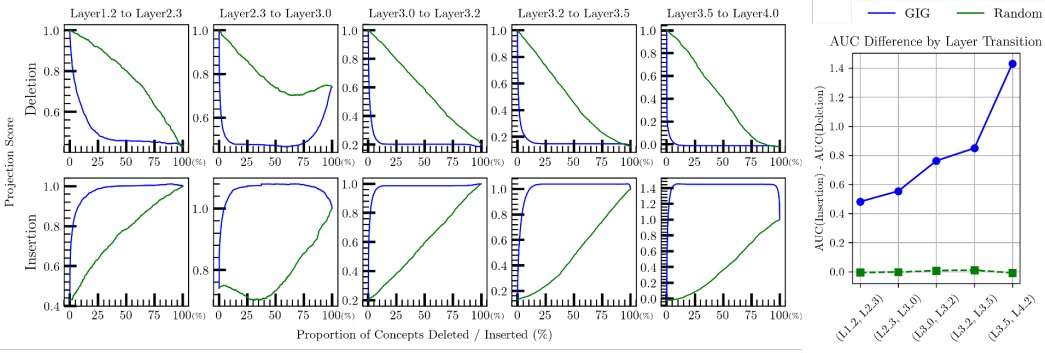

Figure 5: **Left**: Deletion (Top row) and Insertion (Bottom row) scores across consecutive layers in ResNet50. The blue curves represent our method (GIG), while the green curves denote random attribution. Our method consistently outperforms random attribution, as indicated by the significantly steeper decline in deletion scores and the sharper rise in insertion scores. **Right**: AUC Difference by Layer Transition. It quantifies our superiority, showing our AUC difference achieving substantially higher AUC differences across all layer transitions.

the actual region of "dog nose" in the image may constitute only a small portion (e.g, less than 10%) of the total image, removing the most relevant concept from the source layer will likely affect only 1-2 PFVs in the target layer. Therefore, by deleting or inserting concepts in the source layer that most strongly contribute to the "dog nose," and observing the change in the projection magnitude of the one PFV in the target layer that has the largest projection onto the "dog nose" direction, we can determine whether the attribution computed by GIG is valid.

To this end, we plotted the curve of the normalized maximum projection values of the PFVs in the target layer onto the target concept vector direction. Specifically, during the Insertion/Deletion processes, the maximum projection values at each step were normalized by the original maximum projection value prior to any Insertion or Deletion. We refer to this normalized value as the projection score, and this metric as Interlayer Insertion/Deletion.

We conducted the Inter-layer Deletion/Insertion experiments on both GIG attribution and random attribution. For random attribution, we deleted or inserted concept vectors in a random order.

**Results.** The left plot in Fig. 5 displays the Inter-layer Deletion/Insertion curves between various blocks, specifically [Layer1.2, Layer2.3, Layer3.0, Layer3.2, Layer3.5, Layer4.2]. This experiment was conducted on the average projection score of the five most important target layer concepts across 20 random images from the ImageNet validation set. As expected, the curve for GIG attribution shows a rapid decrease/increase during deletion/insertion, outperforming the random attribution.

Interestingly, Insertion curves of GIG sometimes exceed 1, indicating that the insertion of only the positively attributed concepts leads to a higher maximum projection value in the target layer output than that in the original output. The projection score returns to 1, as the original layer output is restored after the negatively attributed concepts are inserted.

The right plot in Fig. 5 shows the difference in AUC between GIG and random attribution. The significant AUC difference in GIG validates our method, demonstrating its effectiveness in accurately attributing the relationship between concept vectors across whole layers.

## 5 CONCLUSION

In this paper, we firstly present a novel approach for extracting and attributing concepts within image models, enhancing interpretability through a comprehensive layer-wise analysis. Unlike existing methods that often confine their explanations to specific classes, our approach provides a comprehensive understanding by analyzing shared concepts throughout the dataset. The shift from class-specific to dataset-wide explanations represents a significant advancement in the field of XAI in image models, allowing for a more holistic understanding of model behavior.

With the dataset of PFV and iERF, we propose a pipeline that systematically decompose PFVs into meaningful concept vectors, and further attribute these concepts across layers using the Generalized Integrated Gradients (GIG) method. With our method, we can reveal how concepts evolve and influence decisions across different layers of the network. Through extensive qualitative and quantitative analyses, we demonstrate the effectiveness of our method in both accurately capturing and utilizing essential features.

Given its potential for broad applicability, we can extend our method to other deep learning architectures, such as Transformer models. Additionally, the implications of analyzing entire datasets rather than focusing solely on class-specific explanations could be more thoroughly investigated. We believe that our approach opens a new avenues for interpretability in image models.

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

## A  ALL-LAYER ANALYSIS

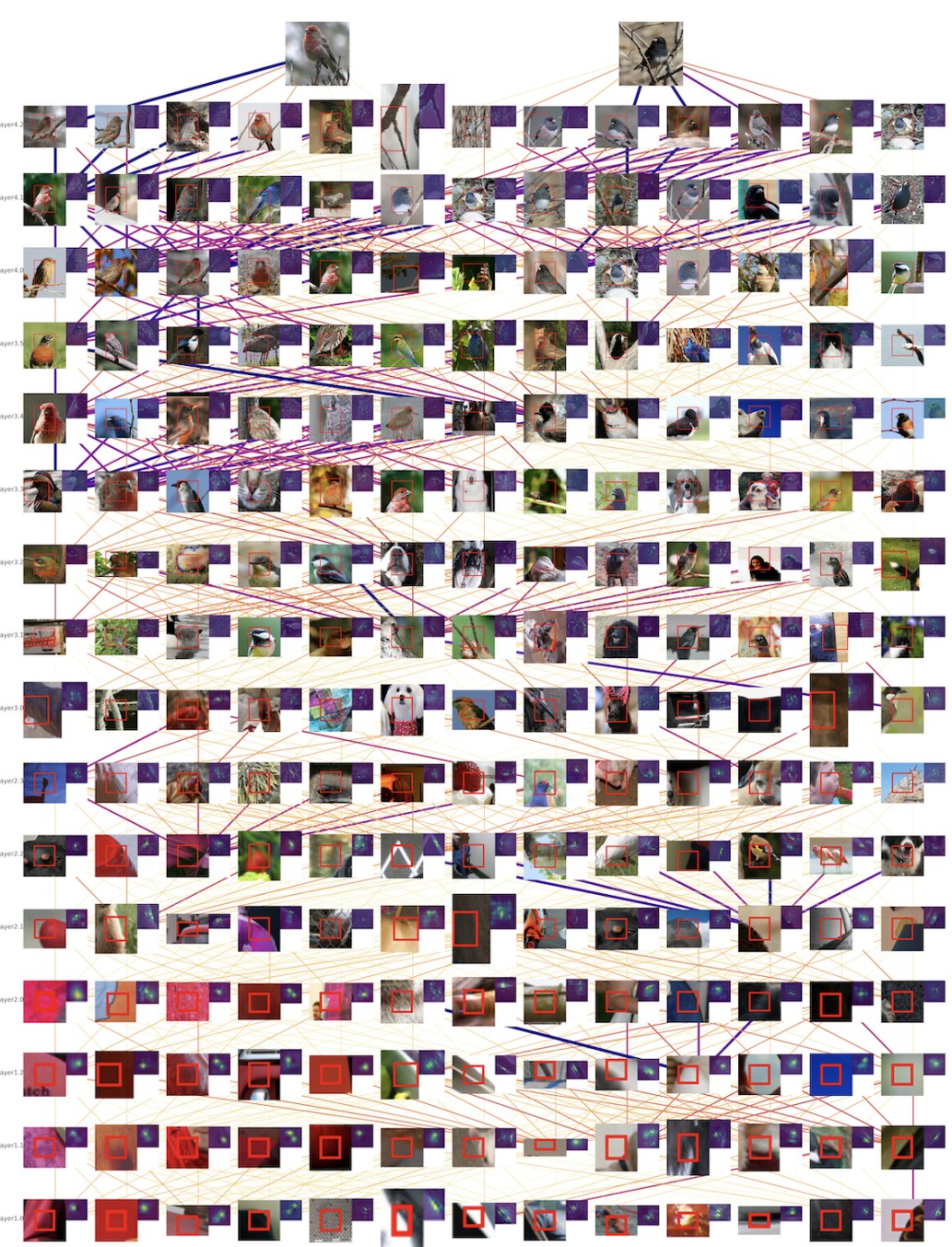

Figure 6: Causal explanation graph of every layer in ResNet50. The top-5 most important concepts in each class and top-3 shared concepts. The thicker and bluer the edge, the stronger the contribution between concepts.

As in Fig. 6, we can decompose every concepts through every layers. More interactive examples can be found at https://iclr2025gig.netlify.app/graph_visualization.html. The GIG scores between layers are normalized to [0,1].

## B    DETAILED PROCEDURE OF CLUSTERING

While traditional k-means clustering methods utilize Euclidean distance, Euclidean distance is not appropriate for high-dimensional data (Huang et al., 2008). Therefore, we employ an angle-based method known as *Spherical k-means* (Dhillon & Modha, 2001). Converting k-means with Euclidean distance to Spherical k-means can be easily achieved by modifying two key steps:

**1. Unit Vector Transformation**: First, we normalize all input vectors $v_i$ to unit vectors. This ensures that all rows of the input matrix $x$ have a norm of 1, focusing on the angular relationship between vectors rather than their magnitude. This is achieved through L2-normalization as follows:

$$\mathbf{v}'_i = \frac{\mathbf{v}_i - \mu_{\text{BN}}}{\|\mathbf{v}_i - \mu_{\text{BN}}\|}$$

Here, $\mathbf{v}_i$ be a PFV. $\mu_{\text{BN}}$ be the normalization mean, which is defined as:

$$\mu_{\text{BN}} = -\frac{\gamma_{\text{BN}}\mu_{rn}}{\sqrt{\sigma_{rn}^2 + \epsilon}} + \beta_{\text{BN}}, \tag{8}$$

where $\gamma_{\text{BN}}$, $\beta_{\text{BN}}$ are the scale, shift parameters, while $\mu_{rn}$ and $\sigma_{rn}$ are the running mean and variance in batch normalization. $\mu_{\text{BN}}$ represents net shift by batch normalization.

**2. Centroid Update with L2 Normalization**: In the traditional k-means algorithm, the centroid of a cluster is computed by taking the sum of the vectors in the cluster and dividing by the number of vectors (L1-normalization). However, in Spherical k-means, we instead compute the sum of the vectors and then apply L2-normalization to ensure that the centroid remains a unit vector. This guarantees that the centroid's magnitude is 1, which is essential for maintaining consistency in the next iteration when calculating distances between centroids and vectors.

The centroid $\mathbf{c}$ of a cluster $\mathcal{C}$ is computed as follows:

$$\mathbf{c} = \frac{\sum_{\mathbf{v}'_i \in \mathcal{C}} \mathbf{v}'_i}{\|\sum_{\mathbf{v}'_i \in \mathcal{C}} \mathbf{v}'_i\|}$$

By adopting these modifications, we ensure that the distance metric used in clustering reflects the angular relationships between data points. In high-dimensional spaces, this is more robust and meaningful than traditional Euclidean distance.

To enhance the clustering process, we adopt the **Bisecting k-means clustering**. This approach begins by treating all the data as a single cluster. In each iteration, the largest cluster is selected for splitting, ensuring that the algorithm focuses on the most substantial portions of the data first, thereby preventing smaller clusters from dominating the early stages of clustering. The selected cluster is then divided into two sub-clusters using the Spherical k-means algorithm. During this process, the vectors are normalized to unit norms, and k-means is applied with $k = 2$, ensuring that the centroids remain unit vectors. After the split, all clusters are re-evaluated, and the next largest cluster is selected for further bisection. This iterative process continues until the desired number of clusters is reached. Once the desired number of clusters is achieved, instead of creating new centroids, we use the average of the vectors within each cluster as the concept vector. By combining Spherical k-means with the bisecting clustering approach, we leverage both methods' strengths: handling high-dimensional data effectively while iteratively refining the cluster structure.

As we approximate the embeddings in the $l$-th layer, $X^l \in \mathbb{R}^{HW \times C}$ with $\tilde{X}^l$, $\tilde{X}^l$ is obtained with

$$\tilde{X}^l = UV^T + \beta, \tag{9}$$

where $\beta$ is $\mu_{\text{BN}}$ in our clustering.

## C    PFV-iERF DATASET PREPARATION

To determine the contribution of each pointwise feature vector (PFV) to the output, any attribution method could be used. In this study, since the dataset of PFV-iERF is from Han et al. (2024), we specifically utilize its approach, Sharing Ratio Decomposition, which distributes the relevance of

| Del/Ins AUC | layer1.0 | layer1.1 | layer1.2 | layer2.0 | layer2.1 | layer2.2 | layer2.3 | layer3.0 |
|---|---|---|---|---|---|---|---|---|
| Bisecting Clustering | 0.0133/0.6039 | 0.0049/0.8982 | 0.0021/0.2827 | 0.0005/0.4856 | 0.0014/0.2827 | 0.0008/0.1809 | 0.0014/0.2807 | 0.0016/0.5400 |
| Dictionary Learning | -/0.3498 | 0.0020/0.3191 | 0.0011/0.2282 | 0.0020/0.4087 | 0.0008/0.2368 | 0.0006/0.0980 | 0.0005/0.0697 | 0.0015/0.6889 |
| Sparse Autoencoder | 0.0011/0.3877 | 0.0002/0.1039 | 0.0001/0.0636 | 0.0003/0.0303 | 0.0001/0.0020 | 0.0001/0.0149 | 0.0004/0.0337 | 0.0003/0.1982 |
|  | layer3.1 | layer3.2 | layer3.3 | layer3.4 | layer3.5 | layer4.0 | layer4.1 | layer4.2 |
| Bisecting Clustering | 0.0008/0.3999 | 0.0017/0.5591 | 0.0017/0.5092 | 0.0013/0.4515 | 0.0008/0.3149 | 0.0022/0.8762 | 0.0029/0.9362 | 0.0016/0.9119 |
| Dictionary Learning | 0.0002/0.2148 | 0.0008/0.2825 | 0.0014/0.3973 | 0.0012/0.3495 | - | - | - | - |
| Sparse Autoencoder | 0.0000/0.0016 | 0.0000/0.0003 | 0.0000/0.0394 | 0.0001/0.0186 | 0.0002/0.0131 | 0.0007/0.9880 | 0.0007 / 0.9860 | 0.0005/0.9809 |

Table 2: Data table for Fig. 4. A lower Deletion AUC indicates better performance, while a higher Insertion AUC is preferable.

every pixel $j$ in layer $k$ back to pixel $i$ in layer $l$ according to the inner product of the respective PFVs, as below:

$$R_i^l = \sum_{j \in PF_i^l} \mu_{i \to j}^{l \to k} R_j^k, \quad \mu_{i \to j}^{l \to k} = \left\langle \frac{\hat{v}_{i \to j}^l}{\|v_j^k\|}, \frac{v_j^k}{\|v_j^k\|} \right\rangle. \tag{10}$$

## D DATA TABLE FOR FIG. 4

Tab. 2 shows the data table to create Fig. 4. In our comparison of bisecting clustering with dictionary learning and sparse autoencoder, we focus on two key metrics: Deletion AUC and Insertion AUC. A lower Deletion AUC indicates better performance, while a higher Insertion AUC is preferable.

## E PFV DECOMPOSITION

As in Sec. 3.2.2, we decompose all PFVs in any layer with the concept vectors. Fig. 7- 9 show the examples of PFV decomposition. For the concept vectors, we present the top-5 most used concept vectors with their top-10 nearest embeddings. And, for the coefficients, we round to the nearest tenth.

## F IMPLEMENTATION DETAILS FOR THE EXPERIMENT

We followed the training setup in Templeton et al. (2024), except for the analysis unit. We analyzed the models with our PFV-iERF dataset, thereby leveraging Sparse AutoEncoder to PFVs instead of tokens.

Given $n$: the input and output dimension, $m$: the autoencoder hidden layer dimension, $s$: the size of the dataset, $W_e \in \mathbb{R}^{m \times n}$: encoder weights, $W_d \in \mathbb{R}^{n \times m}$: decoder weights, $b_e \in \mathbb{R}^m$, $b_d \in \mathbb{R}^n$: biases, the loss function over a dataset $X \in \mathbb{R}^{s,n}$ is as follows:

$$\mathcal{L} = \frac{1}{|X|} \sum_{x \in X} ||x - \hat{x}||_2^2 + \lambda \sum_i |f_i(x)| ||W_{d,i}||_2, \tag{11}$$

where $f(x) = \text{ReLU}(W_e x + b_e)$, and $\hat{x} = W_d f(x) + b_d$.

## G ADDITIONAL QUALITATIVE RESULTS

Fig. 10 shows causal explanation graph of 'Truck-Tractor' and 'English Foxhound-Walkerhound'.

## H COMPUTATION COST ANALYSIS

The GIG method calculates the integrated gradient between layers, specifically, computing the integrated gradient of all concepts at the starting layer with respect to a particular direction at the targeted layer. At first glance, this process appears computationally intensive due to the large number of concepts, denoted as $k$, at the starting layer. For example, layer 3.5 contains 8192 concepts.

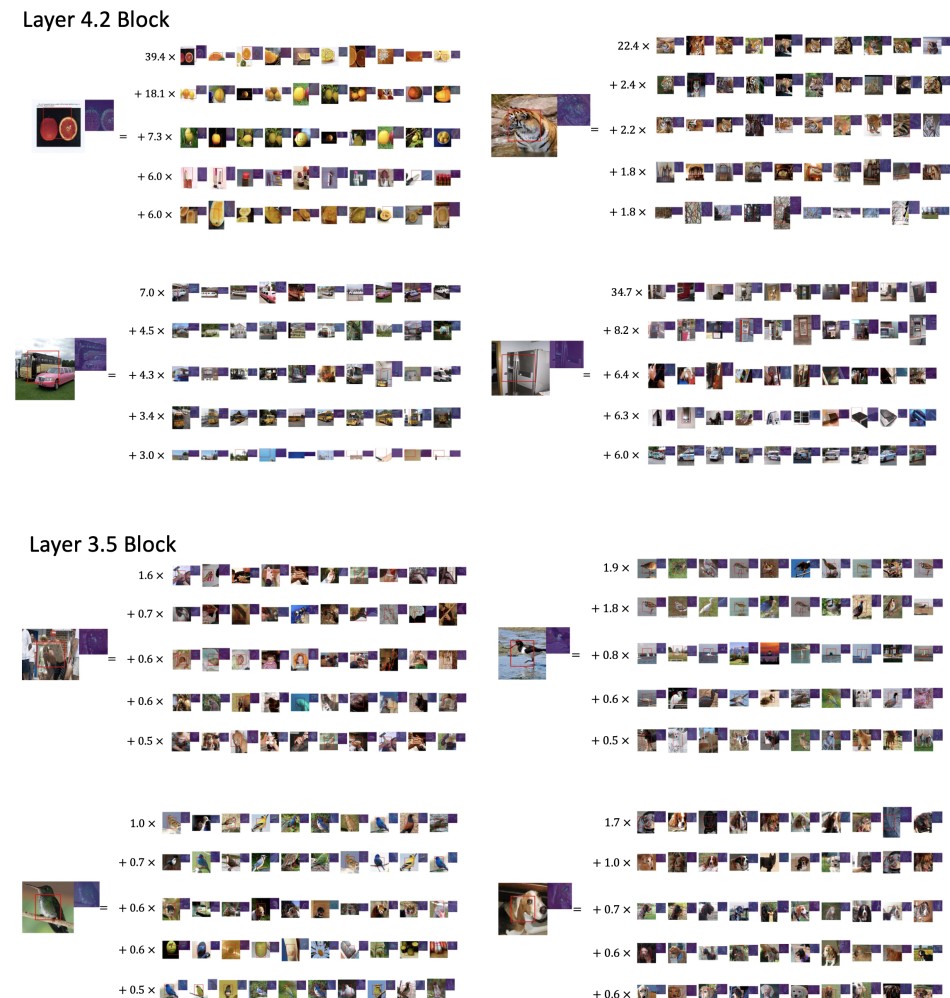

Figure 7: Examples of PFV decomposition in layer4.2 Block (**Top**) and layer3.5 Block (**Bottom**).

The activation of the starting layer, $X^l$, is approximated as $UV^T$, where $U \in \mathbb{R}^{HW \times k}$ and $V \in \mathbb{R}^{C \times k}$. This formulation is computationally equivalent to adding a $1 \times 1$ convolutional layer with weights of size $C \times k$. Since $k$ is set to be 8 times the channel size, calculating $\text{GIG}(\mathbf{v}_q^a \to \mathbf{v}_t^b)$ involves computing this additional convolutional layer.

However, not all concepts are used to construct $UV^T$; in fact, the majority of concepts are unused. Specifically, only a fraction $1 - (l_0 \text{ ratio})$ of the concepts are utilized. For example, in layer 3.5, only 0.38% of the concepts are used, meaning that only 30 out of 8192 concepts are active. As a result, the additional computational cost of this convolutional layer is effectively reduced to the equivalent of adding a convolutional layer with weights of size $C \times (1 - l_0 \text{ ratio})k$.

## I   ABLATION STUDY ON NUMBER OF CONCEPTS

We conducted an ablation study to evaluate the impact of the number of clusters. Specifically, we examined various layers, including [Layer1.2, Layer2.0, Layer2.2, Layer3.0, Layer3,2, Layer 3.5, and Layer4.2]. Also, we evaluated performance across different values of $k$, where $k \in \{0.5 \times n_{channel}, n_{channel}, 2 \times n_{channel}, 4 \times n_{channel}, 8 \times n_{channel}\}$. For example, at Layer3.5, with $n_{channel} = 1024$, we evaluated the following values of $k$ : $[512, 1024, 2048, 4096, 8192]$. The quantitative results are summarized in Table 3.

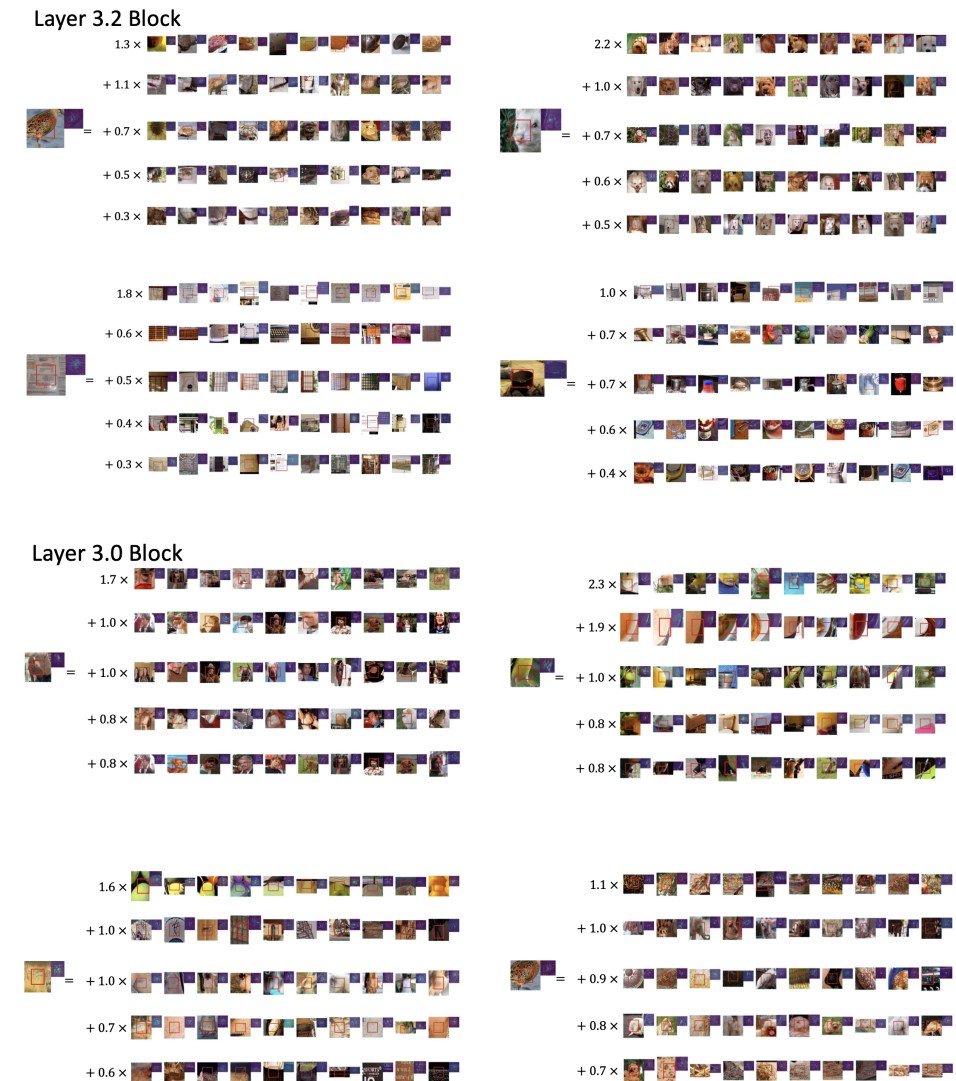

Figure 8: Examples of PFV decomposition in layer3.2 Block (**Top**) and layer3.0 Block (**Bottom**).

| k | Rel $l_2(\downarrow)$ | | | | | $l_0$ ratio($\uparrow$) | | | | |
|---|---|---|---|---|---|---|---|---|---|---|
| | Layer1.2 | Layer2.2 | Layer3.0 | Layer3.5 | Layer4.2 | Layer1.2 | Layer2.2 | Layer3.0 | Layer3.5 | Layer4.2 |
| $0.5 \times n_c$ | 0.5287 | 0.7293 | 0.7052 | 0.6771 | 0.7183 | 0.9542 | 0.9730 | 0.9836 | 0.9834 | 0.9748 |
| $n_c$ | 0.5021 | 0.7030 | 0.6771 | 0.6598 | 0.6758 | 0.9737 | 0.9840 | 0.9901 | 0.9891 | 0.9827 |
| $2 \times n_c$ | 0.4753 | 0.6783 | 0.6447 | 0.6403 | 0.6299 | 0.9856 | 0.9913 | 0.9934 | 0.9929 | 0.9875 |
| $4 \times n_c$ | 0.4544 | 0.6503 | 0.6081 | 0.6181 | 0.5748 | 0.9910 | 0.9940 | 0.9958 | 0.9950 | 0.9910 |
| $8 \times n_c$ | 0.4313 | 0.6177 | 0.5618 | 0.5891 | 0.5095 | 0.9945 | 0.9961 | 0.9971 | 0.9964 | 0.9936 |

Table 3: Performance metrics across different numbers of concepts $k$. As number of concept $k$ increases, the performance increases.

As the number of concepts, $k$, increases, the relative $l_2$ error consistently decreases, indicating enhanced reconstruction performance. Similarly, the $l_0$ sparsity improves with higher values of $k$, showing that the model uses a more fine-grained representation. These results indicate a clear performance improvement as the number of concepts increases, a trend that is further supported by the following qualitative analysis.

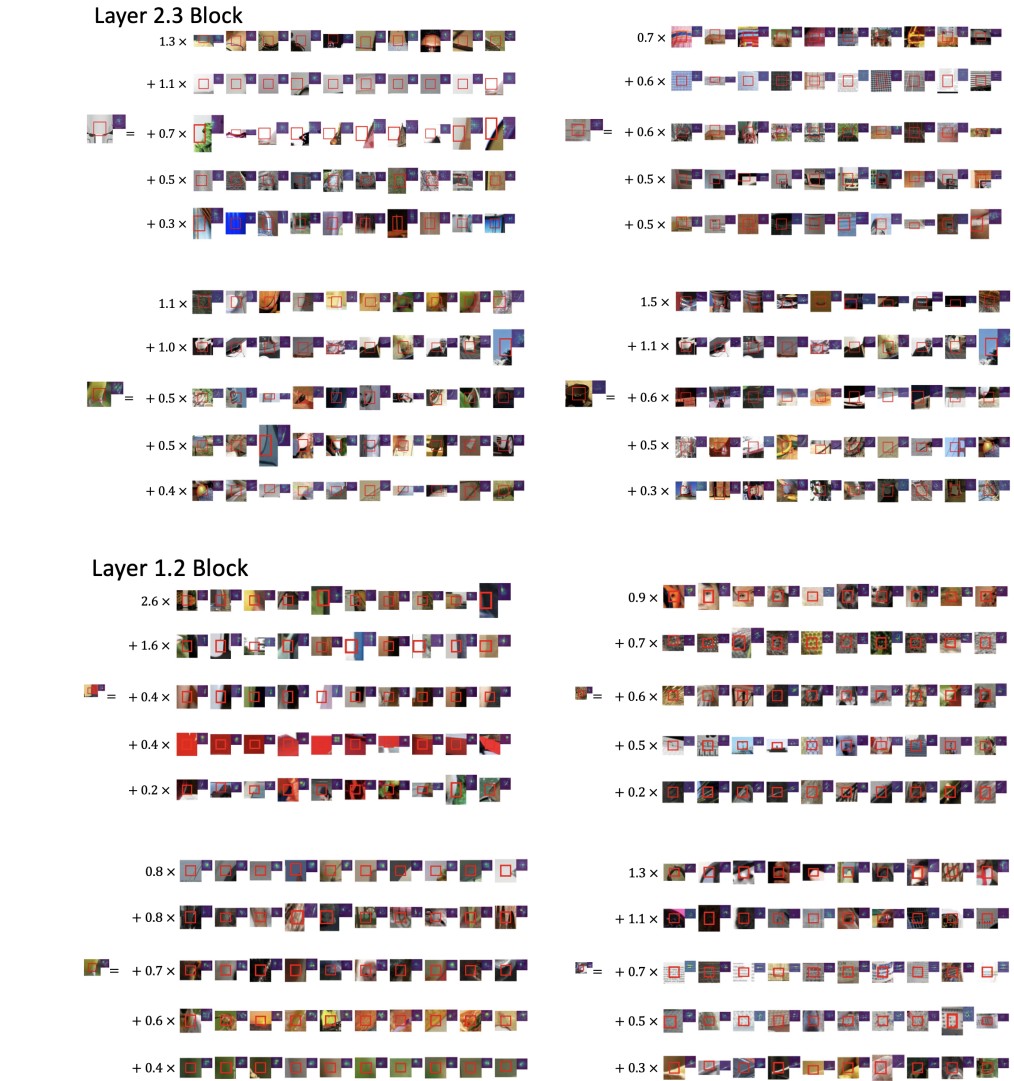

Figure 9: Examples of PFV decomposition in layer2.3 Block (**Top**) and layer1.2 Block (**Bottom**).

Additionally, we conducted a qualitative analysis to explore how the composition and breadth of each concept evolve as the number of concepts, $k$, varies. To do so, we selected two key vectors: the concept vector (center) and the farthest vector from the concept vector within the same cluster. These two vectors were used as defining points to construct a hyperplane, onto which all vectors within the concept cluster were projected. From this setup, we created a trajectory starting from the concept vector (center) towards the farthest vector, extending along this direction. Along the trajectory, we iteratively selected the closest sample to the each point.

In Fig. 11, we provide the concept vector "Dog Ear" and compare the concept clusters at different values of $k$. For $n_{concept} = 512$, the farthest instance appears to be a stone-like horn. At first glance, it might seem unclear why this instance is classified under the concept of "dog ears." However, upon examining the trajectory, we observe that the concept vector of "dog ear" gradually darkens, transforms into goat-like horns, and finally becomes fully horn-like. This indicates that the concept cluster encompasses a spectrum from dog ears to gray horns, grouping them under a single broad concept.

On the other hand, when $n_{concept} = 2048$, the farthest instance is ear-like fluffy fur, indicating that the farthest concept is now closer to the center than before. Additionally, the cosine distance between

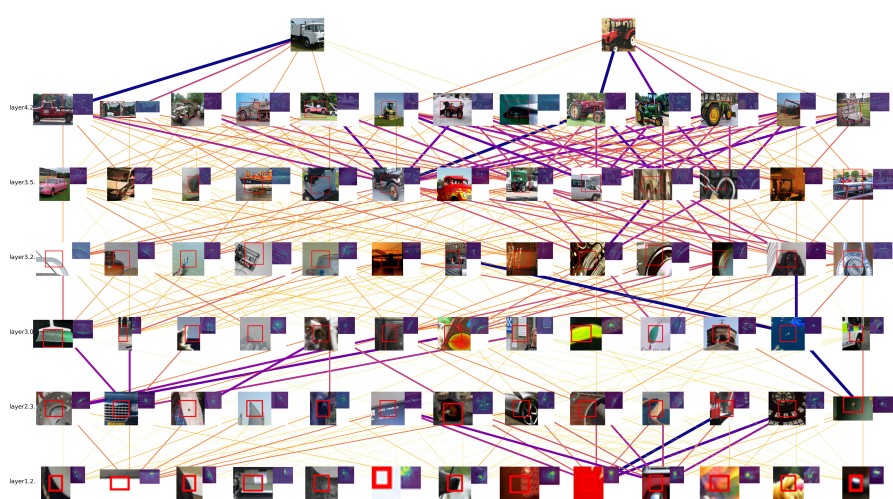

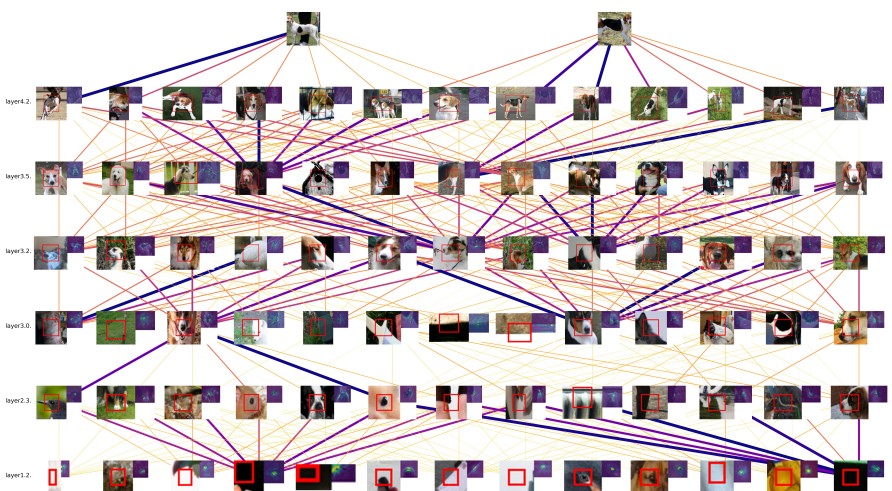

Figure 10: Causal explanation graph of 'Truck-Tractor (**Top**'), and 'English Foxhound-Walkerhound (**Bottom**)'.

the farthest instance and the concept center decreases to $0.6332$, reflecting a narrower range covered by this concept cluster.

Finally, with $n_{concept} = 8192$, the cosine distance further reduces to $0.3801$, indicating a very fine-grained range. In this case, even the farthest concept remains strictly confined to "dog ears" concept, indicating a high degree of specificity.

We found that setting $n_{concept} = 8 \times n_{channel}$ yielded the most balanced results in terms of performance and interpretability. Although performance may continue to improve with higher $K$, we limited our experiments to a maximum of $8192$ concepts. This limitation is due to the fact that our current dataset contains only $50,000$ Effective Receptive Fields (ERFs). With larger values of $k$, the number of ERFs assigned to each concept could fall below one, which would undermine meaningful concept representation.

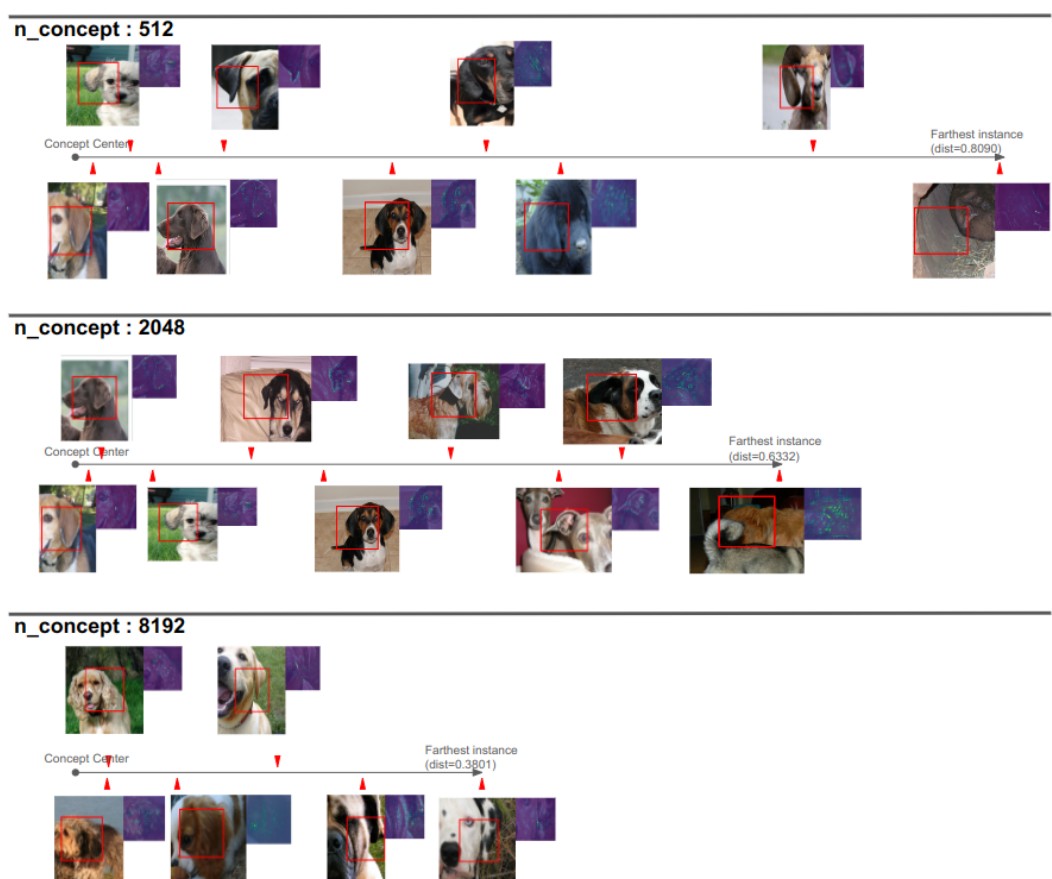

Figure 11: The refinement of the 'dog ear' concept as the number of concepts increases. As the number of concepts grows, the region assigned to each concept becomes finer. **Top**: The trajectory of samples extending towards the farthest instance, "Grey horn," with 512 concepts in Layer 3.5. **Middle**: The trajectory towards "Ear-like tail" with 2048 concepts in Layer 3.5. **Bottom**: The trajectory towards "Dotted dog ear" with 8192 concepts in Layer 3.5.

## J    ROBUSTNESS TESTING FOR IDENTIFIED CONCEPTS

We conducted robustness experiments to evaluate the stability of our concepts. In Fig. 12, we observed that the targeted attacked input (Carlini & Wagner, 2017) was more strongly influenced by the concepts associated with the target class than by those of the original class. Furthermore, we demonstrate how the corrupted concepts are formed. In Fig. 13, we found that the extracted concepts remained consistent even under heavy Gaussian noise. This robustness suggests that our method captures genuinely meaningful patterns that are resilient to input perturbations. Moreover, we show how the noise affects the classification process. The interactive graphs of the attacked samples are available at https://gig2025iclr.netlify.app/graph_visualization_additional.html.

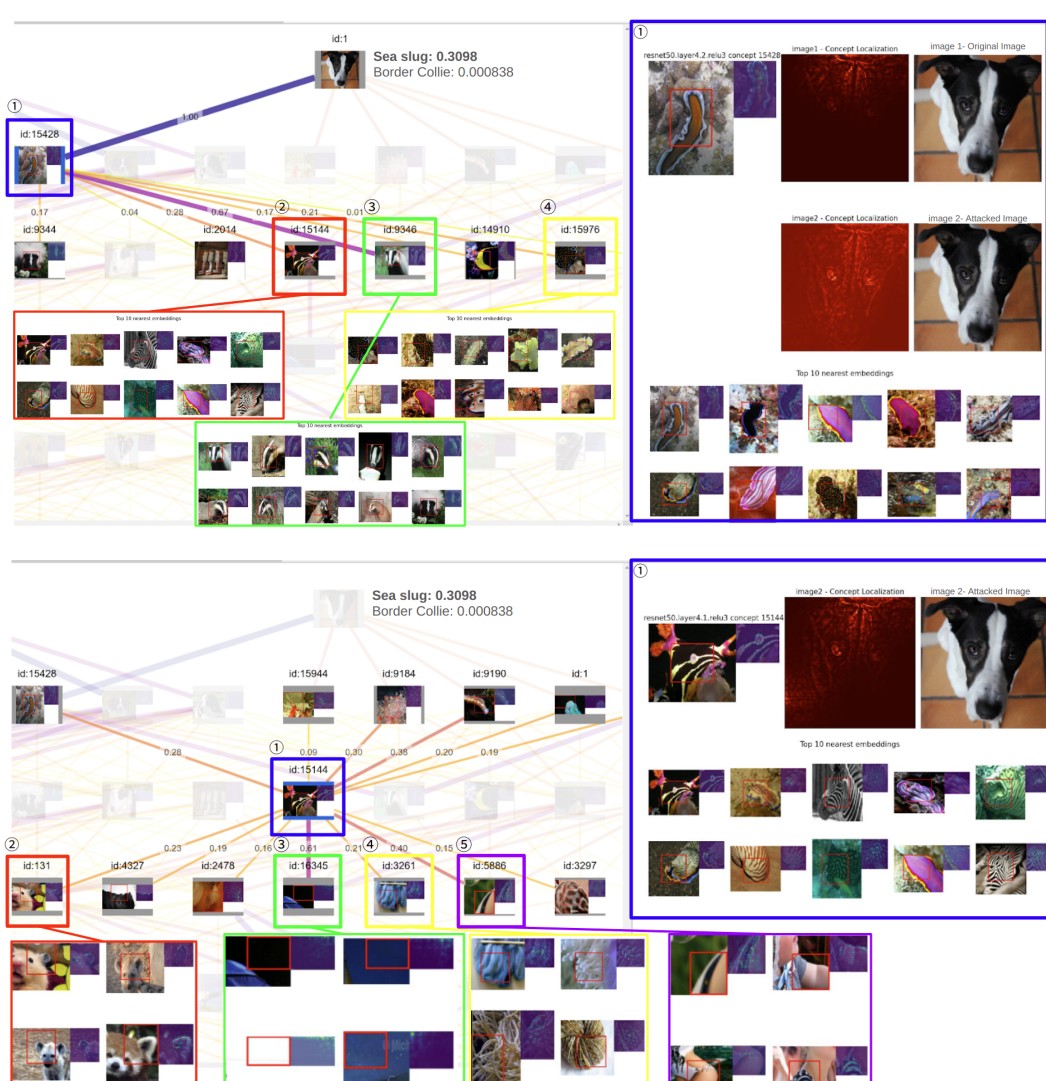

Figure 12: Causal explanation graph for Targeted Adversarial Attack. We attacked the border collie image to be identified as a sea slug. **Top**: The sea slug-related concept (①) was dominant at the last layer. At Layer 4.1, the sea slug concept (①) was most influenced by three key concepts: the stripe concept (②), black-white furry head concept (③), and the slurp body concept (④). Interestingly, the black-white furry head concept, which closely resembles a critical concept used in the correct classification (cosine similarity > 0.9), was also dominant in forming the corrupted higher-level concept of the sea slug. This suggests that the targeted adversarial attack might build corrupted higher-level concepts by combining non-corrupted features with corrupted features. **Bottom**: A similar pattern is evident with the stripe concept (①) at Layer 4.0, which was influenced by the black-white round ear concept (②), the monochrome background corner concept (③), the wrinkle concept (④), and the black-white stripe concept (⑤).

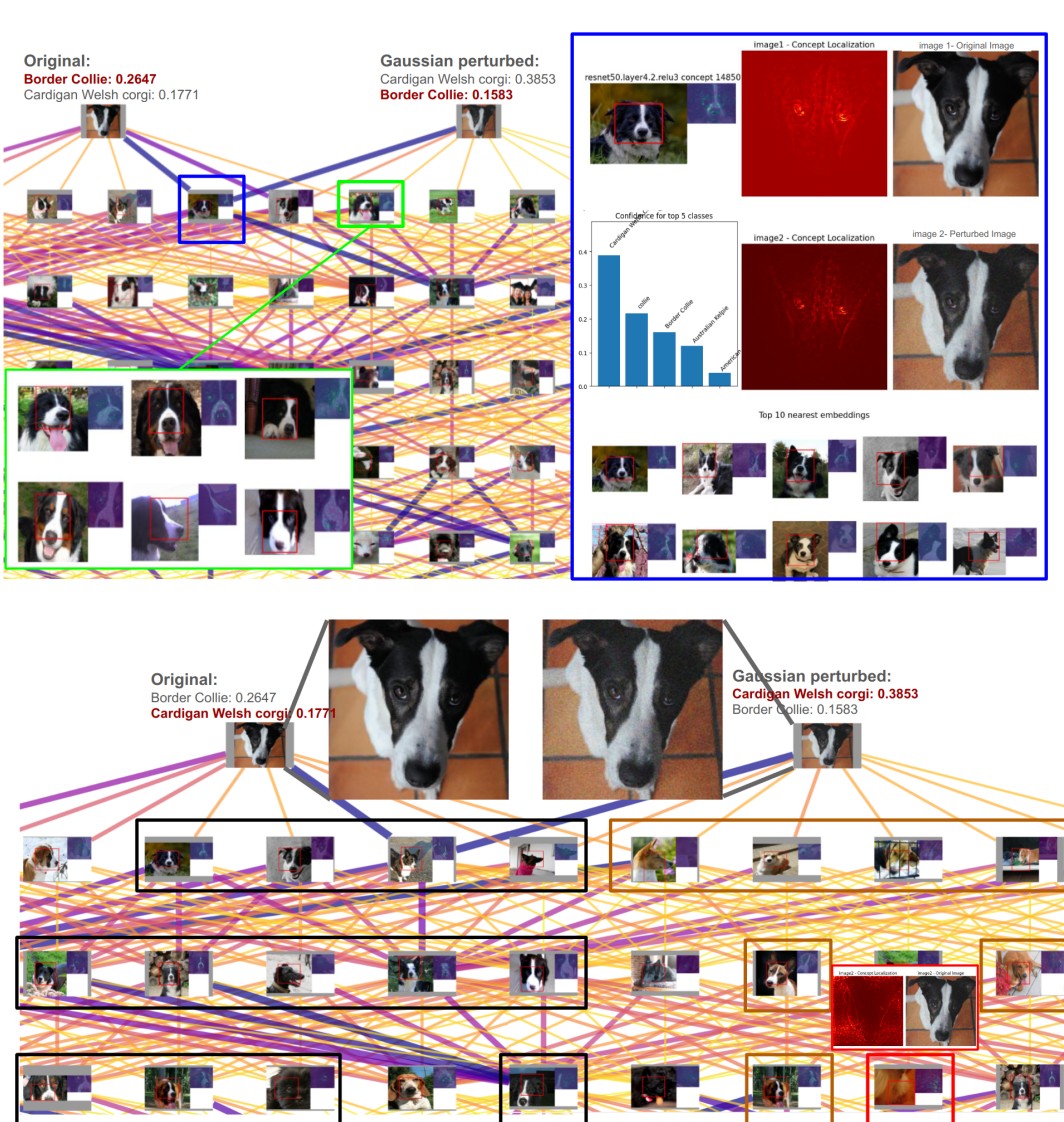

Figure 13: The causal explanation graphs for Gaussian Noise Attack. The model misclassified the border collie image as a Cardigan Welsh corgi due to the heavy Gaussian noise ($\sigma = 70$, considering that RGB values are integers ranging from 0 to 255). **Top**: Causal explanation graph for class 'Border Collie'. Despite the misclassification of the perturbed image, the key concepts essential for correctly identifying the image as a border collie remained intact. This demonstrates the robustness of our method in preserving the underlying causal structure, even under significant noise perturbation. **Bottom**: Causal explanation graph for class 'Cardigan Welsh corgi'. When the image was perturbed with Gaussian noise, brown corgi-related concepts appeared throughout the layers. The graphs revealed that the brown background color played a role in forming the corrupted "brown dog" concept, contributing to the misclassification.

