# OpenReview forum: "Decompose the model: mechanistic interpretability in image models with generalized integrated gradients (GIG)"
_ICLR.cc/2025/Conference — Submitted to ICLR 2025_

### Official Review · Reviewer_N9zG · 2024-10-26

**Soundness:** 3
**Presentation:** 2
**Contribution:** 3
**Rating:** 6
**Confidence:** 5

**Summary:**

This paper proposes an XAI method that first identifies $K$ essential features in each layer, along with their visual cues, using a small image dataset. It then determines the impact of these features on subsequent layers and on the final classification output. This method is model-agnostic and can be applied to both CNNs and ViTs. Rather than providing class-specific explanations, it offers dataset-wide explanations.

**Strengths:**

- They propose a method to analyze the model's features across an entire dataset.
- They provide effective visualizations to convey their insights.
- They use instance-specific Effective Receptive Fields (iERF) to label each identified feature.
- They propose a method to trace a path from input to output, rather than in the reverse direction.
- Their model does not require any additional training.

**Weaknesses:**

- Even with iERF, there is no guarantee of finding a meaningful representation (a human-understandable part across multiple images) for each concept.
- Using ImageNet validation to identify concepts may be limiting, as this dataset may not contain all the features the model has learned during training.
- Some parts of the paper are unclear and difficult to understand, such as the last two sentences of the first paragraph in Section 3.2.1.
- When clustering to find concepts, some may be over-segmented while others may not be segmented at all.
- In Section 4.2.1, you state: "However, as seen in the bottom part of Fig. 4, while the concepts from ‘Ours’ are human-interpretable, those from SAE seem ambiguous and even irrelevant to the class." However, your method may also detect non-meaningful features in some cases.
- Not all identified concepts are meaningful, as seen in Figure 9, where the first row of concepts in the top-left example lacks interpretability.

**Questions:**

- In Section 3.2.1, why did you choose 1 PFV per image-layer? Why not choose 2? What happens to the discovered concepts if this number varies?
- How do you determine the number of clusters? As you know, this has a strong correlation with the dataset being used.
- Why is the claim made in the last paragraph of Section 3.2.2 correct? How do you justify it? What if the discovered concepts are interdependent and unable to span the space where PFVs exist?
- Do the values represented for $l_0$ (ratio) in Table 1 indicate that most of the coefficients are zero and that PFVs are represented by only a few concept vectors?
- In Figure 5, under Deletion, from Layer 2.3 to Layer 3.0, why does the blue line rise at the end?

---

> ### Author Response · Authors · 2024-11-19
>
> ### W1: **Guarantee of Finding Human-Understandable Representations with iERF** & W5/W6: **Detection of Non-meaningful Features**
>
> We acknowledge your concern regarding the guarantee of finding meaningful, human-interpretable representations for each concept using instance-specific Effective Receptive Fields (iERF). While iERF provides a localized understanding of how features contribute to model decisions, it is true that not every identified concept may align with human-defined semantics.
>
> While we acknowledge that some concepts may lack immediate semantic clarity, a substantial portion of the concepts identified by our method are indeed interpretable by humans, as evidenced by the visualizations in our qualitative experiments. These visualizations demonstrate that our method captures shared attributes across multiple classes, which can be intuitively understood by human evaluators.
>
> By focusing on dataset-wide patterns rather than isolated class-specific features, our approach aims to enhance the interpretability of deep models, even if some concepts are abstract or challenging to label. We will continue to refine our method to increase the proportion of human-understandable concepts, particularly by exploring additional evaluation techniques to bridge the gap between model-centric and human-centric interpretations.
>
> ---
>
> ### W2: **Limitations of Using ImageNet Validation Set**
>
> We chose this dataset primarily due to its widespread use and availability, allowing for reproducible and comparable evaluations with other future interpretability methods. However, we recognize that extending our analysis to additional datasets could provide a more comprehensive understanding of the model’s learned representations.
>
> ---
>
> ### W3: **Clarity of Section 3.2.1**
>
> We appreciate the feedback on the clarity of Section 3.2.1. The last two sentences in that section were intended to emphasize our strategy for handling the foreground-background imbalance in images when sampling PFVs. Specifically, we sample one PFV per image based on its contribution to the output logits, thus prioritizing features that are more predictive of the class.
>
> We understand that this explanation could have been clearer, and we will revise this section to improve its readability and precision in the final manuscript.
>
> - **Original**:
>   Thus, we probabilistically sample a single PFV from each image in proportion to its contribution to the output, ensuring a more balanced dataset and solving the imbalance problem.
>
> - **Revised**:
>   Thus, to address the foreground-background imbalance, we sample a single Pointwise Feature Vector (PFV) from each image. This PFV is chosen probabilistically, with a preference for those that contribute more significantly to the model’s output logits. This approach ensures that the selected PFVs are highly relevant to the class predictions, thereby creating a more balanced representation of important features across the dataset.
>
> ---
>
> ### W4: **Over-Segmentation and Under-Segmentation in Clustering** & Q2: **Number of Clusters**
>
> Thank you for pointing out the potential issue of over-segmenting or under-segmenting concepts during clustering. This is indeed a known challenge when applying clustering algorithms to high-dimensional data. To address this, we conducted an ablation study with different numbers of clusters in bisecting k-means. We provide these results in the comment provided to everyone (5) to offer clear guidelines for hyperparameter selection.

---

> > ### Author Response · Authors · 2024-11-19
> >
> > ### Q1: **Why 1 PFV per Image?**
> >
> > We initially experimented with selecting multiple PFVs (e.g., 2 or 3) per image-layer to increase the dataset size and potentially capture more diverse features. However, we found that this approach did not yield a meaningful improvement in the diversity or quality of the discovered concepts.
> >
> > Our sampling strategy is based on the contribution of each PFV to the model's output, with higher weights assigned to features that significantly influence the logits. This probabilistic selection tends to prioritize the most informative regions in each image. When we increased the number of PFVs per image, we observed that the additional PFVs often represented similar regions or features due to the nature of the distribution. Thus, adding more PFVs resulted in redundancy rather than capturing new, distinct concepts.
> >
> > ---
> >
> > ### Q3: **Justification of the Claim in Section 3.2.2**
> >
> > The claim in Section 3.2.2 is based on the assumption that the set of discovered concepts can span the Pointwise Feature Vector (PFV) space, even if the concepts are not entirely independent.
> >
> > - **Overcomplete Representation**:
> >     To address the concern of interdependence among concepts, we intentionally choose the number of clusters $K$ to be larger than the number of channels in each layer. This results in an **overcomplete basis**, which increases the likelihood that the discovered concepts can adequately span the PFV space, even if some of the concepts are correlated or partially redundant. By design, this ensures that our set of concepts covers the entire feature space, preventing gaps in representation.
> >
> > - **Mitigating Redundancy**:
> >     Additionally, we use techniques like **Lasso regularization** during PFV decomposition to promote sparsity and reduce redundancy among selected concepts. This helps in maintaining a diverse set of concepts that still spans the necessary space for accurate model interpretation.
> >
> > We believe this approach provides a balanced trade-off between ensuring coverage and minimizing redundancy.
> >
> > ---
> >
> > ### Q4: **Interpretation of $l_0$ Ratio in Table 1**
> >
> > Yes, the $l_0$ ratio in Table 1 indicates that most coefficients are zero, meaning that PFVs are represented by only a few concept vectors. This high sparsity level is a desirable feature, as it shows that our concept decomposition is efficient, capturing essential features with minimal redundancy. This sparsity also contributes to reducing the computational burden during inter-layer attributions.
> >
> > ---
> >
> > ### Q5: **Explanation of the Blue Line in Figure 5 (Deletion from Layer 2.3 to Layer 3.0)**
> > Thanks to pointing out the anomaly between layer2.3 and layer3.0, this observation led us to uncover an intriguing phenomenon. Now, we are currently conducting experiments and will update the findings promptly as soon as they are available.

---

> ### Author Response · Authors · 2024-11-21
>
> ### **Q5: Explanation of the Blue Line in Figure 5 (Deletion from Layer 2.3 to Layer 3.0)**
> Thanks to pointing out the **anomaly between layer2.3 and layer3.0**, this observation led us to uncover an intriguing phenomenon.Specifically, when we zero out the layer activations (effectively deleting all concepts) and perform a forward pass using only the bias values (i.e., the 3rd batch normalization bias of **layer2.3** and the 1st, 2nd, and 3rd batch normalization biases of **layer3.0**), we observe that the norm of the PFVs at a certain layer (7.72 between **layer2.3** and **layer3.0**) is unexpectedly higher than the norm of the reconstructed vectors(4.64 between **layer2.3** and **layer3.0)**.
>
> In **Figure D-1 (projection)**, we illustrate the ratio of the maximum projection of the concept-deleted PFV to the maximum projection of the original PFV. Mathematically, this is represented as:
>
> $\frac{max(<\Omega',\bar{v}^b_t>)}{max(<\Omega, \bar{v}^b_t>)}$, where $\Omega$ is the output embedding with the complete PFV vectors, $\Omega’$ is the output embedding with the *deleted/inserted* PFV vectors, and $\bar{v}^b_t$ is the normalized version of the target concept vector $v^b_t$.
>
> As $<\cdot, \cdot>$ is the inner product operation, we could rewrite this as:
> \begin{equation}
> \frac{\max(cos(\Omega')\cdot norm(\Omega'))}{\max(cos(\Omega) \cdot norm(\Omega))}
> \end{equation}
> Here, we can rewrite this equation, where $\omega_i$ is the embedding of $\Omega$ at the $i$-th position, and the index $i$ has the $max(\Omega’)$, the index $j$ has the $max(\Omega)$.
> \begin{equation}
> = \frac{cos(\omega'_i)\cdot norm(\omega'_i)}{cos(\omega_j) \cdot norm(\omega_j)}
> = \frac{cos(\omega'_i)}{cos(\omega_j)} \cdot \frac{norm(\omega'_i)}{norm(\omega_j)}
> \end{equation}
> The cosine similarity term, $\frac{cos(\omega'_i)}{cos(\omega_j)}$, shows directional change after the deletion/insertion of the concepts, and the norm ratio term, $\frac{norm(\omega'_i)}{norm(\omega_j)}$, shows the norm ratio changed due to the deletion/insertion.
> We separately plot the cosine similarity term and the norm ratio term in **Figure D-2 (cos)** and **Figure D-3 (norm)**, respectively.
> - **Top:** Deletion experiments
> - **Bottom:** Insertion experiments
>
> Across all layers, the **cosine similarity** either decreases or increases depending on the concept deletion or insertion. However, at specific layers, the **norm ratio** notably increases after concept deletion, resulting in a sharp change in the **projection score**. Especially, during the final 25% of the insertion/deletion process.
>
> In conclusion, these findings demonstrate that concept vectors effectively influence the directions of the **target layer's concept vectors**, as scored by GIG. This observation validates that our method, GIG, accurately measures the importance of concepts by quantifying their contribution to subsequent layers.

---

> > ### Comment · Reviewer_N9zG · 2024-11-25
> >
> > Thank you for your responses. I will keep my score as it is.

---

### Official Review · Reviewer_PsSD · 2024-10-31

**Soundness:** 3
**Presentation:** 3
**Contribution:** 3
**Rating:** 6
**Confidence:** 4

**Summary:**

The paper presents a novel approach to mechanistic interpretability in image models, utilizing Pointwise Feature Vectors (PFVs) and instance-specific Effective Receptive Fields (iERFs) to decompose features into concept vectors, followed by Generalized Integrated Gradients (GIG) for inter-layer concept attribution. The authors focus on dataset-wide analysis in ResNet50, aiming to uncover “shared concepts” that transcend individual classes.

**Strengths:**

+ The combined use of PFVs, iERFs, and GIG offers a comprehensive pathway for analyzing inter-layer relationships in image models.

+ By moving beyond class-specific interpretations, this method identifies cross-class shared concepts, broadening its utility in analyzing dataset-wide patterns.

**Weaknesses:**

- Defining “concepts” as features across channels is ambiguous and possibly arbitrary. The choice of clustering method and number of clusters affects concept formation, raising concerns about consistency and potential cherry-picking.

- The clustering and inter-layer attribution processes, especially GIG, are computationally intensive, limiting the approach’s scalability to larger models and datasets.

- The study primarily focuses on the ResNet50 model, which is convolutional and has a relatively straightforward layer structure. The approach may not directly extend to transformer-based models or architectures with more complex connectivity patterns, limiting its general applicability to other modern architectures.

- Clustering PFVs to form concepts may lead to a loss of granularity in capturing finer details, as individual nuances in features could be overshadowed when grouped into larger concepts. This limitation could make the approach less effective for tasks requiring detailed feature analysis, such as fine-grained object recognition.

- Concept extraction and the interpretability of these concepts can be subjective. Evaluating whether the extracted concepts are meaningful or correctly represent features within the model’s decision-making process is challenging, as human evaluators may have differing opinions on the quality and relevance of these concepts.

- The method does not include robustness testing for the identified concepts, such as how stable they are under input perturbations or across different initializations of the same model. This lack of robustness testing limits the reliability of the concepts for understanding model behavior across varying conditions.

**Questions:**

see Weaknesses

---

> ### Author Response · Authors · 2024-11-19
>
> ### W1: Definition of "Concept" and Choice of clustering method and number of clusters
> **W1-a: Definition of Concept**: Please refer to the comments provided to everyone (2) above for further clarification.
>
> **W1-b: Choice of clustering method and the number of clusters**: We selected bisecting k-means due to its adaptability to the sparse and variably dense nature of PFV data, as discussed in our main text [lines 65-69]. Also, in Sec 4.2.1, we compared our clustering method, Bisecting Clustering, with other concept extraction methods including Dictionary Learning and Sparse AutoEncoder.
>
>   Furthermore, to address your concern on choosing the number of clusters, we conducted additional ablation study, which is Figure B in supplementary materials. For detail, refer to the comment provided to everyone (5).
>
> ---
>
> ### W2: Computational Cost and Scalability GIG.
> We understand the concern regarding the computational cost associated with our method. However, we would like to clarify that the Generalized Integrated Gradients (GIG) approach is optimized for scalability, which also we shared to everyone (4).
>
> ---
>
> ### W3: Generalizability beyond ResNet50.
> Our method is not inherently tied to convolutional architectures. The **PFV-iERF framework** can naturally extend to Vision Transformers (ViTs) by treating each patch as an analysis unit similar to PFVs in convolutional layers. In essence, a convolutional layer can be seen as a fixed-attention Vision Transformer, where each PFV serves as a “token” within the layer. Thus, by replacing PFVs with tokens, our method can be adapted to Transformer-based architectures. We expect that the **interlayer concept attribution**, GIG, also can naturally extend to every models with overcomplete linear bases with their semantically meaningful features.
>
> ---
>
> ### W4: Potential Loss of Granularity in Concept Clustering
> You raised a valid concern about the potential loss of detail when clustering PFVs into broader concepts. We agree that concept aggregation might obscure finer nuances.:
> - **Overcomplete Representation:** We use an overcomplete set of concept vectors (eight times the number of channels), which preserves a high level of detail. This overcompleteness allows us to capture subtle variations within the data. As the ablation study on the number of concepts shows that we can control the fineness of the concept by adjusting $K$, we can increase $K$ for the tasks requiring detailed feature analysis.
>
> ---
>
> ### W5: Subjectivity in Concept Interpretability
> We acknowledge that the interpretability of concepts can be subjective, depending on human evaluators. To address this, we have taken the following steps:
> - **Top-10 Embedding Visualization**: To make the extracted concepts more interpretable and persuadable, we provide the top-10 embeddings with the highest cosine similarity to each concept vector. This helps identifying whether the extracted concepts correctly represent features.
>
> - **Objective Evaluation Metrics**: We have implemented quantitative metrics (C-Deletion, and C-Insertion) to evaluate the relevance and robustness of extracted concepts, reducing reliance on human judgment. These metrics are widely recognized for their ability to validate concept-based explanations across diverse models.
>
> ---
>
> ### W6: Lack of Robustness Testing for Identified Concepts
> Thank you for pointing out the importance of robustness testing for the identified concepts. Your insightful comment inspired us to conduct an additional set of experiments, which turned out to be both interesting and valuable. We are pleased to share the results of these robustness experiments in Comment provided to everyone (6), as we believe they hold enough merit to be shared with every reviewers.
>
> **W6-a: Adversarial Stability**: Please refer to the comment provided to everyone (6).
>
> **W6-b: Cross-Model Consistency**: Also, thank you again for highlighting the importance of evaluating cross-model consistency. In the paper, we used most widely-used pytorch resnet50 (weight=IMAGENET1K_V1) model. And now, we are currently in the process of  conducting on recent version of resnet50 (weight=IMAGENET1K_V2). As both architecture is essentially same, we expect that our method can be applied well. We will update our findings as possible as we can.

---

> > ### Author Response · Authors · 2024-11-24
> >
> > ### **Cross-Model Consistency**:
> > Sorry for the delayed response. We have verified cross-model consistency of our model with different initialization of the same model. The results can be found on the following page: gig2025iclr-imagenetv2.netlify.app/graph_visualization_v2.html.
> >
> > With these experiments including adversarial robustness and cross-model consistency, we can see that our identified concept remains robust.

---

### Official Review · Reviewer_5SYa · 2024-11-03

**Soundness:** 2
**Presentation:** 3
**Contribution:** 2
**Rating:** 3
**Confidence:** 4

**Summary:**

This paper introduces a approach for achieving mechanistic interpretability in image models by systematically tracing data flow through all intermediate layers to the final output across an entire dataset, which borrow from ideas from the language models. The authors propose the use of Pointwise Feature Vectors (PFVs) and Effective Receptive Fields (ERFs) to break down model embeddings into interpretable Concept Vectors. The relevance between these Concept Vectors is quantified using a new technique called Generalized Integrated Gradients (GIG). This methodology provides some analysis of model behavior, moving beyond class-specific explanations and offering a holistic, dataset-wide view of model interpretability. Experiments on ResNet50 demonstrate the effectiveness of this method in both qualitative and quantitative evaluations.

**Strengths:**

1. The introduction of PFVs, ERFs, and the GIG technique represents a viable approach in the field of interpretability for image models. This methodology offers a step towards deeper understanding by focusing on dataset-wide explanations rather than individual class explanations.
2. The authors provide a visualization GUI to help me clearly understand the effectiveness of GIG.

**Weaknesses:**

1. The novelty of this paper is uncertain. It appears to be a straightforward adaptation of mechanistic interpretability analysis from the field of large models, with only minor modifications, such as the introduction of GIG. I recommend that the authors provide a theoretical justification to demonstrate how GIG delivers more accurate and robust interpretability for each concept. By the way, such idea is quite similar to LRP.
2. The paper lacks a clear definition of concept, which is essential for understanding the methodology. Additionally, the definitions of PFV and ERF are somewhat ambiguous, as it is questionable whether DNNs make decisions based solely on individual pixels. In certain models, such as ViTs, decisions may also be patch-based. Clarification on these definitions and their applicability is needed.
3. While the proposed approach seems effective, it may face scalability challenges when applied to large models or datasets due to the computational expense of calculating PFVs and performing clustering. The paper does not adequately address the computational implications of this approach, especially for larger architectures.
4. The paper provides a single example using the ResNet-50 model, which is relatively shallow compared to current large models. To more thoroughly validate the proposed method, additional results using deeper models, such as ViT or CLIP, would be beneficial. Expanding the evaluation to these models could further support the approach’s robustness and generalizability.
5. Minor: most of the figures are not scalable vector graphics.

**Questions:**

1. What is the sensitivity of the method to hyperparameters like the number of clusters in bisecting k-means? It would be helpful to understand how robust the method is to changes in these parameters, as well as any guidelines for selecting optimal values.
2. How does GIG compare in computational performance with other attribution methods? Given the additional complexity introduced by GIG, a comparison of computational efficiency with methods like integrated gradients or SHAP could help users assess the trade-offs of this approach.

---

> ### Author Response · Authors · 2024-11-18
>
> Thank you for recognizing the potential of our proposed PFVs, ERFs, and GIG methodology in advancing the interpretability of image models.
>
> ### W1: **Novelty of our GIG and Theoretical Justification**
>
> This is the first to systematically trace the entire pathway from input image through all intermediate layers to the final output within the whole dataset. Existing mechanistic interpretability analyses in large language models, such as Google DeepMind’s Gemma Scope [6] or Anthropic’s work [7], do not seek interaction between concepts, though it is crucial to explain the model’s prediction mechanistically. Also, they typically treat hidden layer token vectors as input word of their position. However, this is not applicable (and possibly wrong) to the image field, where token vectors (or PFVs) do not represent individual pixels or patches directly. Instead, they correspond to receptive fields, and it is essential to consider instance-specific ERF as the representation of the interested vectors.
>
> Also, we position GIG as part of an evolutionary framework in the field of interpretability, from attribution methods (local explanation methods in Related Works), through concept attribution methods (global), to interlayer concept attribution methods (mechanistic interpretability). Here, we outline the progression from traditional attribution methods to our interlayer concept attribution:
>
> - **Traditional Attribution (e.g., IG, LRP):** Methods like Integrated Gradients (IG) [1] and Layer-wise Relevance Propagation (LRP) [2] have primarily focused on pixel-level or neuron-level attributions within a single layer, explaining model predictions by identifying which input features (pixels) contribute most to the output. These methods work well for instance-specific explanations but often fail to capture higher-level semantic concepts, especially in the context of image models.
>
>
>
> - **Concept Attribution (e.g., TCAV, CAT):** To bridge this gap, concept-based approaches like TCAV [3] and CAT [4] have emerged, focusing on explaining model behavior using high-level concepts rather than individual features. These methods introduced the idea of attributing model predictions to predefined or automatically extracted concepts, but they are typically confined to analyzing a single layer at a time, and also limited to class-specific explanations.
>
>
> - **Interlayer Concept Attribution (our GIG):** Our method advances beyond single-layer concept attribution by introducing interlayer concept attribution. GIG allows us to systematically quantify how concepts evolve and influence model decisions across multiple layers. This is particularly challenging in image models, which are often sparse and highly contextual.
>
> In summary, as our work is the first attempt at dataset-wide mechanistic interpretability to achieve interlayer concept attribution, there are no existing methods for direct comparison as far. We provide an additional evaluation of the robustness of our approach in the
>  comments provided to **everyone** (6).
>
> [1] Sundararajan, M., Taly, A., & Yan, Q. (2017). *Axiomatic Attribution for Deep Networks*. In *Proceedings of the International Conference on Machine Learning (ICML)*, pp. 3319–3328.
>
> [2] Bach, S., Binder, A., Montavon, G., Klauschen, F., Müller, K. R., & Samek, W. (2015). *On Pixel-Wise Explanations for Non-Linear Classifier Decisions by Layer-Wise Relevance Propagation*. *PLOS ONE*, 10(7): e0130140.
>
> [3] Kim, B., Wattenberg, M., Gilmer, J., Cai, C., Wexler, J., Viegas, F., & Sayres, R. (2018). *Interpretability Beyond Feature Attribution: Quantitative Testing with Concept Activation Vectors (TCAV)*. In *Proceedings of the International Conference on Machine Learning (ICML)*, pp. 2668–2677.
>
> [4] Fel, T., et al. (2024). *A Holistic Approach to Unifying Automatic Concept Extraction and Concept Importance Estimation*. In *Advances in Neural Information Processing Systems (NeurIPS)*.
>
> [5] Geva, M., Caciularu, A., Wang, K. R., & Goldberg, Y. (2022). *Transformer Feed-forward Layers Build Predictions by Promoting Concepts in the Vocabulary Space*.
>
> [6] Lieberum, T., et al. *Gemma scope: Open sparse autoencoders everywhere all at once on gemma 2*. *arXiv preprint arXiv:2408.05147* (2024).
>
> [7] Templeton, A. *Scaling monosemanticity: Extracting interpretable features from claude 3 sonnet*. Anthropic, 2024.
>
> ---
>
> For both **W2: Definitions of Concept, PFV, and ERF** and **W3: Scalability and Computational Considerations**,  refer to the comments provided to **everyone**(2-3, and 4, respectively) above for further clarification.
>
> ---

---

> ### Author Response · Authors · 2024-11-18
>
> ---
>
> ### W4: **Evaluation on Larger Models (e.g., ViT, CLIP)**
> We appreciate your interest in the scalability of our method to larger models, such as transformers. However, we would like to clarify that the current study focuses exclusively on convolutional networks, specifically ResNet50, due to its widespread use and well-understood architecture. While transformers present a compelling future direction, the current scope of our research does not include them. We do, however, plan to extend our framework to vision transformers in future work, as our framework is inherently compatible with patch-based architectures such as ViTs.
>
> ---
>
> ### Q1: **Sensitivity to Hyperparameters in Bisecting K-means**
> We agree that hyperparameter sensitivity is an important consideration. To address this, we conducted an ablation study to assess the sensitivity of our method to the number of clusters in bisecting k-means, which is provided on the comments provided to everyone above (5).
>
> ---
>
> ### Q2: **Comparison of GIG with Other Attribution Methods**
> As mentioned in **W1**, our interlayer concept attribution, GIG, differs fundamentally from traditional attribution techniques. While attribution methods like Integrated Gradients (IG) focus on single-layer input-output mappings, GIG captures the hierarchical flow of concepts across layers, providing a more nuanced understanding of the model’s internal mechanisms. Of course, the computational cost of GIG is more expansive than the traditional attribution methods, but we can’t say that this is a trade-off as it offers deeper insights by tracing concept formation across layers.

---

### Official Review · Reviewer_GGFt · 2024-11-03

**Soundness:** 3
**Presentation:** 3
**Contribution:** 3
**Rating:** 5
**Confidence:** 3

**Summary:**

The author proposed an architecture that can trace the pathway from the input to the output across the dataset. Specifically, the author first uses pointwise feature vectors (PFVs) and instance-specific effective receptive fields (iERFs) to generate concept vectors (CVs). Then the author introduces a new method called Generalized Integrated Gradients (GIG) to capture the contribution of a specific concept vector in a layer to both the final output and concept vectors of subsequent layers. Moreover, the author provides extensive experiment results to validate the effectiveness of the proposed method.

**Strengths:**

1. The paper is well-organized and easy to follow.
2. The author provides extensive experiments that can validate the effectiveness of the proposed method.

**Weaknesses:**

1. In Figure 3, the author demonstrates that concept 3 represents the 'rounded cone'. It is better to provide more diverse examples. The current examples are mostly noses which are not sufficient to prove that concept 3 represents the 'rounded cone'. It is better to provide more non-animal cone-shaped examples.

2. For the PFV decomposition, it is better to normalize the total contribution to 1, which can not only help to compare the contribution for each concept but also provide more information when a concept is an inter-class concept. It can provide a better comparison for the inter-class concept. Currently, I cannot fully understand why the decomposition has a result like 39.4 $\times$ concept 1 in the top left of Figure 7, how to understand 39.4 in this example while the concept that contributes most to the result in the bottom left of the example that in layer 4.2 block is only 7?

3. In the Figure 5 Insertion experiment, for the result of layer 3.5 to layer 4, there is a huge drop when inserting the final 5%. However, for the previous layers, there is no such phenomenon. Is there any explanation for this phenomenon?

**Questions:**

N/A

---

> ### Author Response · Authors · 2024-11-18
>
> Thank you for your thoughtful review. We appreciate your positive feedback on the organization, clarity, and extensive experiments that validate the effectiveness of our proposed method. Below is the response to the question you raised. We hope this clarifies any concerns.
>
> ### **W1: Diversity of Examples for Concept 1838 in Figure 3**
>
> We understand your concern regarding the limited diversity of examples for **Concept 1838** ("rounded cone"), as most examples presented were animal noses. To address this, we have included a new figure (**Figure A**) in the supplementary material, which shows the top 100 embeddings of **Concept 1838**. This expanded set of examples illustrates that Concept 1838 is consistently identified as a "rounded cone shape with a black tip," encompassing a broader range of objects beyond animal noses. The diverse examples include various cone-shaped structures, such as certain tools or parts of machinery, which better represent the general characteristics of this concept.
>
> ---
>
> ### **W2: Clarification of Contribution Values and PFV Decomposition**
>
> Before addressing your specific concern, we would like to clarify the distinction between the **Concept Contribution** and the **PFV Decomposition**.
>
> #### **W2-a: Concept Contribution Score** (Normalization Not Required But A Good Option)
>
> The **Concept Contribution Score** evaluates the impact of each concept vector on the classification logits or the subsequent layer’s concept vectors. Specifically, this score quantifies the extent to which a specific concept contributes to the model’s prediction confidence or the formation of more abstract concepts in the following layers. We intend to reflect the contribution score between layers itself as a confidence score, thus we normalize each score with their maximum value of the corresponding class. Yet, we acknowledge that your suggestion, **normalization (sum to one), is also a good option** for comparing inter-class concepts.
>
> #### **W2-b: PFV Decomposition Coefficients** (Absolute Magnitude Difference)
> The **PFV decomposition** is conducted with (overcomplete) linear basis, which is concept vectors in our case. The discrepancy between the norm of top 1 concept vectors, which you observed (e.g., the '39.4' and ‘7’ values in Figure 7), is due to **differences in the magnitude** of comprised PFV vectors. For instance, the PFV magnitude for the first sample (grapefruit) was **101.1618**, while the PFV magnitude for the bus sample was **42.2055**. These substantial differences in the magnitude of PFV vectors directly affect the decomposition coefficients.
>
> ---
>
> ### **W3: Explanation for the Huge Drop in the Insertion Experiment (Figure 5)**
>
> Although we provided an analysis of this phenomenon in the main text [line 513-516], we have added further clarifications.
>
> In earlier layers, the insertion process initially includes positively attributed concepts with the order of their attribution scores of the target concepts. Thus, after most of the concepts are inserted, **negative concepts** are inserted at the last order, which can lead to a slight decrease in the projection values as they may not be directly related to the target class.
> However, as the network progresses to higher layers (Layer 3.5 → Layer 4), the sharp decline observed is due to the insertion of highly specific, high-level concepts.

---

> > ### Comment · Reviewer_GGFt · 2024-12-02
> >
> > Thanks for your reply. After reading the supplementary materials, I still feel confused about the questions. For W1, there are lots of samples don't show like rounded core. For W2, if the magnitude different for different sample, how can we directly understand these numbers, the authors still don't provide enough explanations. Hence, I will decrease my rate to 5.

---

> > > ### Author Response · Authors · 2024-12-03
> > >
> > > We apologize for not providing sufficient explanation earlier, which may have caused confusion.
> > >
> > > Firstly, for **W1**, concept “rounded cone” is a label we assigned based on our observation of the top 10 embeddings. Thus, it may not seem a rounded cone for someone. Yet, examining the top 10, or even the top 100, embeddings clearly suggests that they share a common concept within the concept cluster.
> > > By focusing on dataset-wide patterns rather than isolated class-specific features, we aim to enhance the interpretability of deep models, even if **some concepts are abstract** or challenging to label. We will continue to refine our method to increase the proportion of human-understandable concepts, particularly by exploring additional evaluation techniques to bridge the gap between model-centric and human-centric interpretations.
> > >
> > > And for **W2**, the PFV (Pointwise Feature Vector) value represents the aggregation of **activations within a specific layer**, and these values naturally vary across samples due to the unique characteristics of each input. Since our concept vectors are **unit vectors**, differences in the coefficient values are expected, as they reflect the varying degrees of alignment between the activations and the concept vector for each sample.
> > >
> > > We hope this explanation adequately addresses your concerns and resolves any confusion. Please let us know if further clarification is needed.

---

### Author Response · Authors · 2024-11-18
**To everyone**

Dear reviewers,

We sincerely thank you for your constructive feedback and thoughtful reviews of our submission. We appreciate the positive remarks on the novelty and clarity of our work, as well as your insightful suggestions for improvement. Below, we provide a response addressing key points raised and emphasize important aspects of our contributions:

---

### 1. **Emphasis on Our Main Contributions**

We appreciate the recognition of our work’s strengths and would like to reiterate the key contributions of our paper clearly:

- We first introduce a **novel framework for interlayer mechanistic interpretability** in image models, systematically decomposing pointwise feature vectors (PFVs) into interpretable concept vectors **across all layers within the whole dataset**, which goes beyond both layer-specific and class-specific explanations seen in prior works.

- Additionally, to **quantify the interlayer causal contributions** of specific concept vectors, we propose an interlayer attribution method called **Generalized Integrated Gradients (GIG)**, which extends the attribution method of Integrated Gradients (IG).

These contributions advance our understanding of the internal mechanisms of networks, providing deeper insights into how complex image models process data.

---

### 2. **Clarification of the Definition of 'Concept’**

We have noted your request for a more precise definition of **'concept'** in our framework. In our framework, a concept is defined as a linear basis in the high-dimensional PFV space that represents semantically meaningful features across different images, transcending  classes. For example, concepts can represent shared visual patterns like "striped texture" or "rounded shapes" that appear across multiple classes. These concepts are discovered via bisecting k-means clustering of PFVs, where each concept is both interpretable and dataset-wide.

---

### 3. **Explanations of Pointwise Feature Vector (PFV) and Instance-specific Effective Receptive Field (iERF)**

To aid in understanding, we would like to reiterate the key concepts central to our work.

- **Pointwise Feature Vector (PFV)**: A PFV is a feature vector along the channel axis extracted from a specific spatial location within a hidden layer of the network, effectively summarizing the information contained within the receptive field at that location.

- **Instance-specific Effective Receptive Field (iERF)**: The concept of iERF extends the traditional notion of the receptive field (RF), which represents the region of the input image that contributes to the activation of a particular neuron or feature vector in the network. The theoretical receptive field often covers a large area, sometimes even exceeding the size of the input image, especially in deeper layers (e.g., in Layer 3, the RF can be larger than the entire image itself).

  To address this, we define the **instance-specific Effective Receptive Field (iERF)** as the most relevant subset of the receptive field that significantly impacts the activation of a given PFV. Unlike the standard RF, which is determined purely by the architecture's kernel sizes and strides, the iERF focuses on regions that actually influence the PFV’s activation. We achieve this by applying attribution methods (e.g., **Sharing Ratio Decomposition, Integrated Gradients**) to determine the specific pixels or patches within the receptive field that contribute most strongly to the PFV’s activation.

  The instance-specific Effective Receptive Field (iERF) serves as a visual label of the corresponding PFV, indicating the input region that most significantly influences the PFV’s activation. This approach provides a more intuitive interpretation of the underlying semantic features.
---

### 4. **Computational Cost of Generalized Integrated Gradients (GIG)**

We understand the concerns regarding the computational cost of **GIG**, particularly for larger models. We will discuss about this point in the revised manuscript. However, it is important to highlight that the computational overhead of GIG is significantly reduced due to its efficient design. Specifically, the number of computations is determined by:

- The **number of concepts**
- Multiplied by the **(1 - $l_0$ ratio)** (sparsity of the PFV decomposition)
- And the **attribution computation** between layers

This results in a much smaller computational load. For example, even though there are **8192 concepts at layer 3.4**, the model only uses **32 concepts**, so we can ignore computation of the remaining **8160 concepts**.

---

---

> ### Author Response · Authors · 2024-11-18
>
> ### 5. **Ablation Study on Number of Concepts ($K$)**
> We conducted an ablation study to assess the impact of the number of clusters. Here, we focused on a single layer (**Layer 3.5**) with ($n_{channel} = 1024$). We evaluated performance across various numbers of concepts, $(K = [512, 1024, 2048, 4096, 8192])$. The quantitative results are summarized in the **Table 1**.
>
>   Table 1: Performance Metrics Across Different Numbers of Concepts
> | $n_{concept}$ | $\textrm{Rel}~l_2 (\downarrow) $ | $ l_0~\textrm{ratio} (\uparrow)$ |
> | --- | --- | --- |
> | 512 | 0.6771 | 0.9834 |
> | 1024 | 0.6598 | 0.9891 |
> | 2048 | 0.6403 | 0.9929 |
> | 4096 | 0.6181 | 0.9950 |
> | 8192 | 0.5891 | 0.9964 |
>
> As the number of concepts, $K$, increases, the relative $l_2$ error consistently decreases, indicating better reconstruction performance. Also, the $l_0$ sparsity improves with higher $K$, showing that the model uses a more fine-grained representation. These results indicate a clear improvement in performance with an increasing number of concepts, which is also seen in the following qualitative analysis.
>
> Additionally, we performed **qualitative analysis** to assess how broadly each concept applies as the number of concepts, K, varies.  First, we selected two key vectors: the **concept vector (center)** and the **farthest vector** from the concept vector within the concept cluster. Using these two vectors as bases, we constructed a hyperplane. We then projected all vectors within the concept cluster onto this plane. From this setup, we drew a trajectory starting from the concept center towards the farthest vector, extending along this direction. As the trajectory extended, we iteratively selected the closest sample to the current point on the trajectory.
>
> In **Figure B** in the supplementary materials, we provide a concept vector “Dog Ear” and compare the concept clusters of varying number of concepts, $K$.  With $n_{concept} = 512$, the farthest instance appears as a stone-like horn. At first glance, it might be unclear why this instance is categrozed under the concept of “dog ears.” However, by examining the trajectory, we observe that the concept vector of dog ear gradually darken, transform into goat-like horns, and finally become fully horn-like. This indicates that the concept cluster encompasses a spectrum from ears to gray horns, grouping them under a single broad concept.
>
> On the other hand, when $n_{concept}$ is $2048$, the farthest instance is a ear-like fluffy fur, indicating that the farthest concept is closer to the center than before. Additionally, the cosine distance decreases to 0.6332, reflecting a narrower range covered by this concept cluster.
>
> Finally, with $n_{concept} = 8192$, the cosine distance further reduces to 0.3801, indicating a very fine-grained range. In this case, even the farthest concept remains strictly confined to “dog ears.”
>
> We found that setting of $n_{concept}=8 \times n_{channel}$ yielded the most balanced results in terms of performance and interpretability. Although performance may continue to improve with higher $K$, we limited our experiments to a maximum of 8192 concepts. This limitation is due to the fact that our current dataset contains only 50,000 Effective Receptive Fields (ERFs), resulting in cases where the number of ERFs assigned to each concept could fall below one, which would undermine meaningful concept representation.

---

> ### Author Response · Authors · 2024-11-19
>
> ### 6. **Robustness Testing for Identified Concepts**
>   We have performed robustness experiments to assess the stability of our concepts:
> **1) Targeted Adversarial Attack** [8]:We observed the targeted attacked input is more influenced by the concepts of the target class than those of the original class. Furthermore, we showed how the corrupted concepts formed.
> **2) Gaussian Noise Perturbation**:  We found that the extracted concepts remained consistent, even under challenging conditions. This robustness suggests that our method captures genuinely meaningful patterns that are not easily disrupted by input noise. Moreover, our method can show how noise influenced the classification.
>
> For detail, please refer to the **[C_Robustness.pdf]** in the supplementary materials. Also, we will update our interactive page for this results as soon as we are ready.
> With these two experiments, we demonstrate the robustness of our method in explaining the adversarial attacks (which often fails with the traditional local, attribution methods) and preserving the underlying causal structure even with noise perturbation.
>
> [8] Carlini, Nicholas, and David Wagner. "Towards evaluating the robustness of neural networks." 2017 ieee symposium on security and privacy (sp). Ieee, 2017.

---

> > ### Author Response · Authors · 2024-11-21
> >
> > With the robustness testing results, we updated our project page on https://gig2025iclr.netlify.app/graph_visualization_additional.html.

---

> > > ### Author Response · Authors · 2024-11-25
> > >
> > > We have carefully incorporated all of your suggestions into our revised manuscript and have submitted it for your review. We have revised both the main manuscript and the appendix, with all changes marked in blue for your convinience.

---

### Meta-Review · Area_Chair_jNoX · 2024-12-15

**Metareview:**

This paper proposes a novel approach to model interpretation by tracing the entire pathway from the input, through all intermediate layers, to the final output. Additionally, a concept decomposition method is introduced to analyze the local focus of the model. Despite its innovative ideas, the paper received mixed reviews, with a tendency toward negative assessments.

While the proposed concepts aim to enhance interpretability, the approach remains largely sample-specific, making it challenging to identify class-specific or shared concepts that could provide more general insights into the model's decision-making process. This limitation hinders human understanding of the underlying mechanisms. The idea of tracing the entire pathway is intriguing, but its robustness is uncertain. The potential influence of small-weighted connections on sensitive features is not thoroughly addressed, raising concerns about the approach's reliability. Moreover, the lack of comparisons with other prototype-based models limits the contextual evaluation of the proposed method.

**Additional Comments On Reviewer Discussion:**

Some parts are not clear, especially the concepts are not human understandable.

---

### Decision · Program_Chairs · 2025-01-22

Reject